# A general approximation lower bound in $L^p$ norm, with applications to feed-forward neural networks

**El Mehdi Achour**[1], **Armand Foucault**[1], **Sébastien Gerchinovitz**[2,1], **and François Malgouyres**[1]

[1]Institut de Mathématiques de Toulouse ; UMR5219 , Université de Toulouse ; CNRS , UPS IMT
F-31062 Toulouse Cedex 9, France
[2]IRT Saint Exupéry, 3 rue Tarfaya, 31400 Toulouse, France

```
{El_mehdi.achour,armand.foucault,francois.malgouyres} AT
                  math.univ-toulouse.fr
    sebastien.gerchinovitz AT irt-saintexupery.com
```

## Abstract

We study the fundamental limits to the expressive power of neural networks. Given two sets $F$, $G$ of real-valued functions, we first prove a general lower bound on how well functions in $F$ can be approximated in $L^p(\mu)$ norm by functions in $G$, for any $p \geq 1$ and any probability measure $\mu$. The lower bound depends on the packing number of $F$, the range of $F$, and the fat-shattering dimension of $G$. We then instantiate this bound to the case where $G$ corresponds to a piecewise-polynomial feed-forward neural network, and describe in details the application to two sets $F$: Hölder balls and multivariate monotonic functions. Beside matching (known or new) upper bounds up to log factors, our lower bounds shed some light on the similarities or differences between approximation in $L^p$ norm or in sup norm, solving an open question by DeVore et al. [DHP21]. Our proof strategy differs from the sup norm case and uses a key probability result of Mendelson [Men02].

## 1 Introduction

Neural networks are known for their great expressive power: in classification, they can interpolate arbitrary labels [ZBH+21], while in regression they have universal approximation properties [Cyb89, Hor91, LLPS93, KL20], with approximation rates that can outperform those of linear approximation methods [Yar18, DHP21]. Though the approximation problem is often only one part of the underlying learning problem (where generalization and optimization properties are also at stake), understanding the fundamental limits to the approximation properties of neural networks is key, both conceptually and for practical issues such as designing the right network architecture for the right problem.

**Setting and related works.** One way to quantify the expressive power of neural networks is through the following problem (some informal statements will be made more precise in the next sections). Let $G$ be the set of all functions $g_{\mathbf{w}} : \mathcal{X} \subset \mathbb{R}^d \to \mathbb{R}$ that can be represented by tuning the weights $\mathbf{w} \in \mathbb{R}^W$ of a feed-forward neural network with a fixed architecture, and let $F$ be any set of real-valued functions on $\mathcal{X}$. A natural question is: how well can functions $f \in F$ be approximated by functions $g_{\mathbf{w}} \in G$? More precisely, given a norm $\|\cdot\|$ on functions, what is the order of magnitude of the (worst-case) *approximation error of $F$ by $G$* defined by

$$\sup_{f \in F} \inf_{g_{\mathbf{w}} \in G} \|f - g_{\mathbf{w}}\| \,, \tag{1}$$

and how small can it be given the numbers $W$, $L$ of weights and layers, and some properties of $F$?

36th Conference on Neural Information Processing Systems (NeurIPS 2022).

Lower bounds on the approximation error (1) can be useful in several ways. They provide a limit to the best approximation accuracy that one can hope to achieve if the number of weights or layers of the network is constrained, and help design optimal architectures under these constraints. They also imply a lower bound on the minimal number of weights or layers to include in a network in order to approximate any function in $F$ with a given accuracy $\varepsilon$.

The case when $\|\cdot\|$ is the sup norm (defined by $\|f\|_\infty = \sup_{x\in\mathcal{X}}|f(x)|$) is rather well understood at least in some special cases. For example, when $F$ is a Hölder ball of smoothness $s > 0$ (a.k.a. Hölder exponent) and the network uses the ReLU activation function, Yarotsky [Yar17] derived a lower bound on (1) of the order of $W^{-2s/d}$, later refined to $(LW)^{-s/d}$ (up to log factors) by [Yar18, YZ20] when the depth of the network varies from $L = 1$ to $L \approx W$. Using the bit extraction technique, these authors showed that these lower bounds are achievable (up to log factors) with a carefully designed ReLU network architecture. Refined results in terms of width and depth were obtained by [SYZ22] when $s \leq 1$, while some other activation functions were also studied in [YZ20].

In this paper, we study (1) with the $L^p(\mu)$ norm, defined by $\|f\|_{L^p(\mu)} = \left(\int_X |f(x)|^p d\mu(x)\right)^{1/p}$, for $1 \leq p < +\infty$ and some probability measure $\mu$ on $\mathcal{X}$. There is a qualitative difference between measuring the error in sup norm or in $L^p(\mu)$ norm, $p < +\infty$. In the former case, the error is small only if the approximation is good over the whole domain. In the latter case, the error can be small even if the approximation is inaccurate over a small portion of the domain. Since the $L^p(\mu)$ approximation problem corresponds to approximating functions in $F$ in a more "average" sense than in sup norm, a natural question is whether the same accuracy can be achieved with a smaller network or not. Unfortunately, however, the proof strategies behind the lower bounds of [Yar17, Yar18, YZ20, SYZ22] are specific to the sup norm (see Remark 1 in Section 3 for details). DeVore et al. [DHP21] indeed commented: "When we move to the case $p < \infty$, the situation is even less clear [...] we cannot use the VC dimension theory for $L^p(\Omega)$ approximation. [...] What is missing vis-à-vis Problem 8.13 is what the best bounds are and how we prove lower bounds for approximation rates in $L^p(\Omega)$, $p \neq \infty$."

**Existing lower bounds in $L^p(\mu)$ norm.** Several papers provided lower bounds in some special cases, under some restrictions on the set to approximate $F$, the neural network, the approximation metric, or the encoding map $f \in F \mapsto \mathbf{w}(f) \in \mathbb{R}^W$.

When $F$ is a space of smoothness $s$, a first result which is based on [DHM89] states that when imposing the weights to depend continuously on the function to be approximated, one can not achieve a better approximation rate than $W^{-\frac{s}{d}}$.

For the same $F$, another result for $p = 2$ and for activation functions which are continuous ([Mai99, MMR99]) proves a lower bound on the approximation of functions of smoothness $s$ on a compact of $\mathbb{R}^d$, by one hidden-layer neural networks, of order $W^{-\frac{s}{d-1}}$. A matching upper bound is proven for a particular activation function, which is sigmoidal but pathological ([MP99]). For this same activation function, they prove that contrary to the one-hidden-layer case, there is no lower bound in the case of two-hidden-layer networks. The result is based on the Kolmogorov-Arnold superposition theorem.

In [SX21], the authors study approximation by shallow neural networks with bounded weights and activations of the form ReLU$^k$ for an integer $k$. They approximate the closure of the convex hull of shallow ReLU$^k$-neural networks with constrained weights. They obtain optimal lower bounds of order $W^{-\frac{1}{2}-\frac{2k+1}{2d}}$ in any norm $\|\cdot\|_X$, where $X$ is a Banach space to which the approximation functions belong and such that these functions are uniformly bounded w.r.t. $\|\cdot\|_X$. Although we only consider approximation in $L^p(\mu)$ norm, our results complement the latter by addressing neural networks with unbounded weights and arbitrary depth, and general sets $F$.

Approximation lower bounds in $L^p(\mu)$ norm, $p \geq 1$, have also been studied in the quantized neural networks setting (networks with weights encoded with a fixed number of bits). In [PV18], under weak assumptions on the activation function, the authors prove a lower bound on the minimal number of nonzero weights $W$ that are required for a network to approximate a class of binary classifiers with $L^p$ error at most $\varepsilon$. They show that $W$ is at least of the order $\varepsilon^{-\frac{p(d-1)}{\beta}} \log_2^{-1}(1/\varepsilon)$, where $\beta$ is a smoothness parameter. Later works including [VP19, GR20] derive lower bounds for approximation by quantized networks in various norms.

**Main contributions and outline of the paper.** We prove lower bounds on the approximation error (1) in any $L^p(\mu)$ norm, for non-quantized networks of arbitrary depth, and general sets $F$. Our main contributions are the following.

In Section 2, we first prove a general lower bound for any two sets $F$, $G$ of real-valued functions on a set $\mathcal{X}$ (Theorem 1). The lower bound depends on the packing number of $F$, the range of $F$, and the fat-shattering dimension of $G$. We then derive a versatile corollary when $G$ corresponds to a piecewise-polynomial feed-forward neural network (Corollary 1), solving the question by DeVore et al. [DHP21]. Importantly, our proof strategy still relies on VC dimension theory, but differs from the sup norm case in using a key probability result of Mendelson [Men02], to relate approximation in $L^p(\mu)$ norm with the fat-shattering dimension of $G$.

In Sections 3–4 we apply this corollary to the approximation of two sets: Hölder balls and multivariate monotonic functions. Beside matching (known or new) upper bounds up to log factors, our lower bounds shed some light on the similarities or differences between approximation in $L^p$ norm or in sup norm. In particular, with ReLU networks, Hölder balls are not easier to approximate in $L^p$ norm than in sup norm. On the contrary, the approximation rate for multivariate monotonic functions depends on $p$. In Section 5, we outline several other examples of function sets $F$ and $G$ for which the general lower bound (Theorem 1) can also be easily applied. Finally, some proofs are postponed to the supplement, while some details on other existing lower bound proof strategies are provided in the supplement, in Appendix C.

**Additional bibliographical remarks** There are many other related results that we did not mention to keep the focus on our specific approximation problem. For instance, depth separation results show that deep neural networks can approximate functions that cannot be as easily approximated by shallower networks (e.g., [Tel16, VRPS21]). Let us also mention the general results of [YB99], which characterize minimax rates of estimation based on metric entropy conditions. Understanding the precise connections between these statistical results and our general approximation lower bound is an interesting question for the future.

**Definitions and notation.** We provide below some definitions and notation that will be used throughout the paper. We denote the set of positive integers $\{1, 2, \ldots\}$ by $\mathbb{N}^*$ and let $\mathbb{N} := \mathbb{N}^* \cup \{0\}$. All sets considered in this paper will be assumed to be nonempty. We will not explicitly mention $\sigma$-algebras; for instance, by "Let $\mathcal{X}$ be a measurable space" we mean that $\mathcal{X}$ is a set implicitly endowed with a $\sigma$-algebra.

Let $p \in [1, +\infty]$ and $\mathcal{X}$ be any measurable space endowed with a probability measure $\mu$. For any measurable function $f : \mathcal{X} \to \mathbb{R}$, the $L^p(\mu)$ norm of $f$ is defined by $\|f\|_{L^p(\mu)} = \left( \int_{\mathcal{X}} |f(x)|^p d\mu(x) \right)^{1/p}$ (possibly infinite) if $p < +\infty$, and $\|f\|_{L^\infty(\mu)} = \text{ess sup}_{x \in \mathcal{X}} |f(x)|$. We will write $\lambda$ for the Lebesgue measure on $[0, 1]^d$.

For any $\varepsilon > 0$, two functions $f_1$, $f_2$ are said to be $\varepsilon$-*distant* in $\| \cdot \|$ if $\|f_1 - f_2\| > \varepsilon$. Let $F$ be a set of functions from $\mathcal{X}$ to $\mathbb{R}$. A set $\{f_1, \ldots, f_N\} \subset F$ is said to be an $\varepsilon$-packing of $F$ in $\| \cdot \|$ (or just an $\varepsilon$-packing for short) if for any $i \neq j \in \{1, \ldots, N\}$, $f_i$ and $f_j$ are $\varepsilon$-distant in $\| \cdot \|$. The $\varepsilon$-packing number $M(\varepsilon, F, \| \cdot \|)$ is the largest cardinality of $\varepsilon$-packings (possibly infinite).

For $\gamma > 0$, we say that a set $S = \{x_1 \ldots, x_N\} \subset \mathcal{X}$ is $\gamma$-*shattered* by $F$ if there exists $r : S \to \mathbb{R}$ such that for any $E \subset S$, there exists $f \in F$ satisfying for all $i = 1, \ldots, N$, $f(x_i) \geq r(x_i) + \gamma$ if $x_i \in E$, and $f(x_i) \leq r(x_i) - \gamma$ if $x_i \notin E$. The $\gamma$-*fat-shattering dimension of $F$*, denoted by $\text{fat}_\gamma(F)$, is the largest number $N \geq 1$ for which there exists $S \subset \mathcal{X}$ of cardinality $N$ that is $\gamma$-shattered by $F$ (by convention, $\text{fat}_\gamma(F) = 0$ if no such set $S$ exists, while $\text{fat}_\gamma(F) = +\infty$ if there exist sets $S$ of unbounded cardinality $N$). Similarly, we say that $S$ is *pseudo-shattered* by $F$ if there exists $r : S \to \mathbb{R}$ such that for any $E \subset S$, there exists $f \in F$ satisfying for all $i = 1, \ldots, N$, $f(x_i) \geq r(x_i)$ if $x_i \in E$, and $f(x_i) < r_i$ if $x_i \notin E$. The *pseudo-dimension* $\text{Pdim}(F)$ is the largest number $N \geq 1$ for which there exists $S \subset \mathcal{X}$ of cardinality $N$ that is pseudo-shattered by $F$ (same conventions).[1]

A formal definition of feed-forward neural networks is recalled in Appendix A. In short, in this paper, a *feed-forward neural network architecture* $\mathcal{A}$ of depth $L \geq 1$ is a directed acyclic graph with $d \geq 1$ input neurons, $L - 1$ hidden layers (if $L \geq 2$), and an output layer with only one neuron. Skip connections are allowed, i.e., there can be connections between non-consecutive layers. Given an

---

[1] By definition, note that $\gamma \mapsto \text{fat}_\gamma(F)$ is non-increasing and that $\text{fat}_\gamma(F) \leq \text{Pdim}(F)$ for all $\gamma > 0$.

activation function $\sigma : \mathbb{R} \to \mathbb{R}$, a feed-forward neural network architecture $\mathcal{A}$, and a vector $\mathbf{w} \in \mathbb{R}^W$ of weights assigned to all edges and non-input neurons (linear coefficients and biases), the network computes a function $g_\mathbf{w} : \mathbb{R}^d \to \mathbb{R}$ defined by recursively computing affine transformations for each hidden or output neuron, and then applying the activation function $\sigma$ for hidden neurons only (see Appendix A for more details). Finally, we define $H_\mathcal{A} := \{g_\mathbf{w} : \mathbf{w} \in \mathbb{R}^W\}$ to be the set of all functions that can be represented by tuning all the weights assigned to the network.

A function $\sigma : \mathbb{R} \to \mathbb{R}$ is *piecewise-polynomial* on $K \geq 2$ pieces, with maximal degree $\nu \in \mathbb{N}$, if there exists a partition $I_1, \ldots, I_K$ of $\mathbb{R}$ into $K$ nonempty intervals, such that $\sigma$ restricted on each $I_j$ is polynomial with degree at most $\nu$ (in particular, $\sigma$ can be discontinuous).

## 2 A general approximation lower bound in $L^p(\mu)$ norm

In this section, we provide our two main results: a general lower bound on the $L^p(\mu)$ approximation error of $F$ by $G$, i.e., $\sup_{f \in F} \inf_{g \in G} \|f - g\|_{L^p(\mu)}$, and a corollary when $G$ corresponds to a feed-forward neural network with a piecewise-polynomial activation function. The weak assumptions on $F$ make the last result applicable to a wide range of cases of interest, as shown in Sections 3–5.

### 2.1 Main results

Our generic lower bound reads as follows, and is proved in Section 2.2. We follow the conventions $0 \times \log^2(0) = 0$ and $P^{-\frac{1}{\alpha}} \log^{-\frac{2}{\alpha}}(P) = +\infty$ when $P = 1$.

**Theorem 1.** *Let $1 \leq p < +\infty$ and $\mathcal{X}$ be a measurable space endowed with a probability measure $\mu$. Let $F, G$ be two sets of measurable functions from $\mathcal{X}$ to $\mathbb{R}$, such that all functions in $F$ have the same range $[a, b]$ for some $a < b$, and such that $\mathrm{fat}_\gamma(G) < +\infty$ for all $\gamma > 0$. Then, there exists a constant $c > 0$ depending only on $p$ such that*

$$\sup_{f \in F} \inf_{g \in G} \|f - g\|_{L^p(\mu)} \geq \inf \left\{ \varepsilon > 0 : \log M\big(3\varepsilon, F, \|\cdot\|_{L^p(\mu)}\big) \leq c\, \mathrm{fat}_{\frac{\varepsilon}{32}}(G) \log^2\left(\frac{2\,\mathrm{fat}_{\frac{\varepsilon}{32}}(G)}{\varepsilon/(b-a)}\right) \right\}. \tag{2}$$

*In particular, if $\log M\big(\varepsilon, F, \|\cdot\|_{L^p(\mu)}\big) \geq c_0 \varepsilon^{-\alpha}$ for some $c_0, \varepsilon_0, \alpha > 0$ and all $\varepsilon \leq \varepsilon_0$, and if $\mathrm{Pdim}(G) < +\infty$, then there exist constants $c_1, \varepsilon_1 > 0$ depending only on $b - a$, $p$, $c_0$, $\varepsilon_0$ and $\alpha$ such that*

$$\sup_{f \in F} \inf_{g \in G} \|f - g\|_{L^p(\mu)} \geq \min \left\{ \varepsilon_1,\ c_1 \mathrm{Pdim}(G)^{-\frac{1}{\alpha}} \log^{-\frac{2}{\alpha}}\big(\mathrm{Pdim}(G)\big) \right\}. \tag{3}$$

The first lower bound (2) is generic but requires solving an inequation.[2] In (3) we solve this inequation when $\log M\big(\varepsilon, F, \|\cdot\|_{L^p(\mu)}\big)$ grows at least polynomially in $1/\varepsilon$ (which is typical of nonparametric sets) and when $G$ has finite pseudo-dimension $\mathrm{Pdim}(G)$. Though we will restrict our attention to such cases in all subsequent sections, we stress that the first bound should have broader applications. A first example is when $\mathrm{Pdim}(G) = +\infty$ but $\mathrm{fat}_\gamma(G) < +\infty$ for all $\gamma > 0$ (e.g., for RKHS [Bel18]). The first bound should also be useful to prove (slightly) tighter lower bounds when $\log M\big(\varepsilon, F, \|\cdot\|_{L^p(\mu)}\big)$ has a (slightly) different dependency on $1/\varepsilon$ (e.g., of the order of $\varepsilon^{-\alpha} \log^\beta(1/\varepsilon)$ as when $F$ is the set of all multivariate cumulative distribution functions [BGL07]).

In the rest of the paper, we focus on the important special case when the approximation set $G$ is the set $H_\mathcal{A}$ of all real-valued functions that can be represented by tuning the weights of a feed-forward neural network with fixed architecture $\mathcal{A}$ and a piecewise-polynomial activation function. By combining Theorem 1 with known bounds on the pseudo-dimension [BHLM19], we obtain the following corollary, which bounds the approximation error in terms of the number $W$ of weights and the depth $L$ (i.e., the number of hidden and output layers). The proof is postponed to Appendix B.4.

---

[2]Note that any $\varepsilon \geq (b-a)/3$ is a solution to this inequation, since $\log M\big(3\varepsilon, F, \|\cdot\|_{L^p(\mu)}\big) = \log(1) = 0$ (because all functions in $F$ are $[a, b]$-valued) and $c\, \mathrm{fat}_{\frac{\varepsilon}{32}}(G) \geq 0$. Therefore, the right-hand side of (2) is at most $(b-a)/3$.

**Corollary 1.** *Let $1 \leq p < +\infty$, $d \geq 1$ and $\mathcal{X}$ be a measurable subset of $\mathbb{R}^d$ endowed with a probability measure $\mu$. Let $F$ be a set of measurable functions from $\mathcal{X}$ to $[a, b]$ (for some real numbers $a < b$), such that $\log M\big(\varepsilon, F, \|\cdot\|_{L^p(\mu)}\big) \geq c_0 \varepsilon^{-\alpha}$ for some $c_0, \varepsilon_0, \alpha > 0$ and all $\varepsilon \leq \varepsilon_0$.*

*Let $\sigma : \mathbb{R} \to \mathbb{R}$ be any piecewise-polynomial activation function of maximal degree $\nu \in \mathbb{N}$ on $K \geq 2$ pieces. Then, there exist $W_{\min} \in \mathbb{N}^*$ and $c_1, c_2, c_3 > 0$ such that, for any $W \geq W_{\min}$, any $L \geq 1$, and any fixed feed-forward neural network architecture $\mathcal{A}$ of depth $L$ with $W$ weights, the set $H_{\mathcal{A}}$ of all real-valued functions on $\mathcal{X}$ that can be represented by the network (cf. Section 1) satisfies*

$$\sup_{f \in F} \inf_{g \in H_{\mathcal{A}}} \|f - g\|_{L^p(\mu)} \geq \begin{cases} c_1 W^{-\frac{2}{\alpha}} \log^{-\frac{2}{\alpha}}(W) & \text{if } \nu \geq 2, \\ c_2 (LW)^{-\frac{1}{\alpha}} \log^{-\frac{3}{\alpha}}(W) & \text{if } \nu = 1, \\ c_3 W^{-\frac{1}{\alpha}} \log^{-\frac{3}{\alpha}}(W) & \text{if } \nu = 0. \end{cases} \tag{4}$$

There are equivalent ways to write the above corollary. For example, given a target accuracy $\varepsilon > 0$ and a depth $L \geq 1$, (4) yields a lower bound on the minimum number $W$ of weights that are needed to get $\sup_{f \in F} \inf_{g \in H_{\mathcal{A}}} \|f - g\|_{L^p(\mu)} \leq \varepsilon$. Some earlier approximation results were written this way (e.g., [Yar17, PV18]).

## 2.2 Proof of Theorem 1

In order to prove Theorem 1, we need two inequalities. The first one is straightforward (and appeared within proofs, e.g., in [YZ20]), but formalizes the key idea that if $G$ approximates $F$ with error $\varepsilon$, then $G$ has to be at least as large as $F$. We use the conventions $\log(+\infty) = +\infty$ and $+\infty \leq +\infty$.

**Lemma 1.** *Let $p \geq 1$ and $\mathcal{X}$ be a measurable space endowed with a probability measure $\mu$. Let $F$, $G$ be two sets of measurable functions from $\mathcal{X}$ to $\mathbb{R}$. If $\sup_{f \in F} \inf_{g \in G} \|f - g\|_{L^p(\mu)} < \varepsilon$, then*

$$\log M\big(3\varepsilon, F, \|\cdot\|_{L^p(\mu)}\big) \leq \log M\big(\varepsilon, G, \|\cdot\|_{L^p(\mu)}\big) .$$

*Proof.* Let $P_F = \{f_1, \ldots, f_N\}$ be a $3\varepsilon$-packing of $F$, with $N \geq 1$. Let $P_G = \{g_1, \ldots, g_N\}$ be a subset of $G$ such that $\|f_i - g_i\|_{L^p(\mu)} \leq \varepsilon$ for all $i$. Note that the existence of such a $P_G$ is guaranteed by the assumption $\sup_{f \in F} \inf_{g \in G} \|f - g\|_{L^p(\mu)} < \varepsilon$. Since the $f_i$'s are pairwise $3\varepsilon$-distant in $L^p(\mu)$, the triangle inequality entails that the $g_i$'s are also at least pairwise $\varepsilon$-distant in $L^p(\mu)$. Therefore, $P_G$ is an $\varepsilon$-packing of $G$, and the result follows. $\qquad\square$

The next inequality is a fundamental probability result due to Mendelson [Men02]. It bounds from above the $\varepsilon$-packing number in $L^p(\mu)$ norm of any uniformly bounded function set in terms of its fat-shattering dimension. Crucially, the inequality holds for finite $p \geq 1$, as opposed to the lower bound strategy of Yarotsky [Yar17, Yar18] (see also [DHP21]), that relates the VC-dimension with the approximation error in sup norm. The next statement is a slight generalization of a result of [Men02] initially stated for $[a, b] = [0, 1]$ and for Glivenko-Cantelli classes $G$ (see Appendix B.1 for details).

**Proposition 1** ([Men02], Corollary 3.12). *Let $G$ be a set of measurable functions from a measurable space $\mathcal{X}$ to $[a, b]$ (for some real numbers $a < b$), and such that $\mathrm{fat}_\gamma(G) < +\infty$ for all $\gamma > 0$. Then for any $1 \leq p < +\infty$, there exists $c > 0$ depending only on $p$ such that for every probability measure $\mu$ on $\mathcal{X}$ and every $\varepsilon > 0$,*

$$\log M\big(\varepsilon, G, \|\cdot\|_{L^p(\mu)}\big) \leq c\, \mathrm{fat}_{\frac{\varepsilon}{32}}(G) \log^2\left(\frac{2(b-a)\,\mathrm{fat}_{\frac{\varepsilon}{32}}(G)}{\varepsilon}\right) . \tag{5}$$

Refinements of this inequality were proved in specific cases such as the $L^2(\mu)$ norm [MV03] (see also [Gue17] for empirical $L^p(\mu_n)$ norms). However, using the result of [MV03] when $p = 2$ would only yield a minor logarithmic improvement in the lower bound of Theorem 1.

*Proof (of Theorem 1).* **Part 1.** We start by proving (2), using Proposition 1 as a key argument. Since functions in $G$ are not necessarily uniformly bounded, we will apply Proposition 1 to the "clipped version of $G$". More precisely, for any function $g \in G$, we define its clipping (truncature) to $[a, b]$ as the function $\tilde{g} : \mathcal{X} \to \mathbb{R}$ given by $\tilde{g}(x) = \min(\max(a, g(x)), b)$ for all $x \in \mathcal{X}$. We then set $G_{[a,b]} = \{\tilde{g} : g \in G\}$, which by construction consists of functions that are all $[a, b]$-valued.

Noting that clipping can only help since elements of $F$ are $[a,b]$-valued (see Lemma 4 in the supplement, Appendix B.2), we have

$$\sup_{f \in F} \inf_{g \in G} \|f - g\|_{L^p(\mu)} \geq \sup_{f \in F} \inf_{\tilde{g} \in G_{[a,b]}} \|f - \tilde{g}\|_{L^p(\mu)} \ . \tag{6}$$

Setting $\Delta := \sup_{f \in F} \inf_{\tilde{g} \in G_{[a,b]}} \|f - \tilde{g}\|_{L^p(\mu)}$, we now show that $\Delta$ is bounded from below by the right-hand side of (2). To that end, it suffices to show that every $\varepsilon > \Delta$ is a solution to the inequation

$$\log M\big(3\varepsilon, F, \|\cdot\|_{L^p(\mu)}\big) \leq c \operatorname{fat}_{\frac{\varepsilon}{32}}(G) \log^2 \left( \frac{2(b-a) \operatorname{fat}_{\frac{\varepsilon}{32}}(G)}{\varepsilon} \right) \ . \tag{7}$$

The last inequality is true whenever $\varepsilon \geq (b-a)/3$ (see Footnote 2). We only need to prove (7) when $\Delta < \varepsilon < (b-a)/3$. In this case, by definition of $\Delta$ and by Lemma 1 applied to $G_{[a,b]}$, we have

$$\begin{aligned}
\log M\big(3\varepsilon, F, \|\cdot\|_{L^p(\mu)}\big) &\leq \log M\big(\varepsilon, G_{[a,b]}, \|\cdot\|_{L^p(\mu)}\big) \\
&\leq c \operatorname{fat}_{\frac{\varepsilon}{32}}(G_{[a,b]}) \log^2 \left( \frac{2(b-a) \operatorname{fat}_{\frac{\varepsilon}{32}}(G_{[a,b]})}{\varepsilon} \right) \\
&\leq c \operatorname{fat}_{\frac{\varepsilon}{32}}(G) \log^2 \left( \frac{2(b-a) \operatorname{fat}_{\frac{\varepsilon}{32}}(G)}{\varepsilon} \right) \ ,
\end{aligned} \tag{8}$$

where the second inequality follows from Proposition 1 (note from Lemma 3 in the supplement, Appendix B.2 that $\operatorname{fat}_\gamma(G_{[a,b]}) \leq \operatorname{fat}_\gamma(G)$ for all $\gamma > 0$, which is finite by assumption), and where (8) follows from the next remark. Either $\operatorname{fat}_{\frac{\varepsilon}{32}}(G_{[a,b]}) = 0$, and (8) is true by the convention $0 \times \log^2(0) = 0$ and $c \operatorname{fat}_{\frac{\varepsilon}{32}}(G) \geq 0$. Either $\operatorname{fat}_{\frac{\varepsilon}{32}}(G_{[a,b]}) \geq 1$, and (8) follows from $t \mapsto ct \log^2\left(\frac{2(b-a)t}{\varepsilon}\right)$ being non-decreasing on $[\varepsilon/(2(b-a)), +\infty)$ and $\varepsilon/(2(b-a)) \leq 1/6 \leq 1 \leq \operatorname{fat}_{\frac{\varepsilon}{32}}(G_{[a,b]}) \leq \operatorname{fat}_{\frac{\varepsilon}{32}}(G)$. To conclude, every $\varepsilon > \Delta$ satisfies (7), which implies that $\Delta$ is bounded from below by the right-hand side of (2). Combining with (6) concludes the proof of (2).

**Part 2.** Set $\varepsilon_1' = \min\big\{\frac{\varepsilon_0}{3}, 2(b-a)\big\}$. We now derive (3) from (2). To that end, setting $P = \operatorname{Pdim}(G)$, we show that every $\varepsilon > 0$ satisfying (7) is such that $\varepsilon \geq \min\big\{\varepsilon_1, \ c_1 P^{-\frac{1}{\alpha}} \log^{-\frac{2}{\alpha}}(P)\big\}$, where $\varepsilon_1 \in (0, \varepsilon_1']$ and $c_1 > 0$ will be defined later. Since the claimed lower bound on $\varepsilon$ is true when $\varepsilon \geq \varepsilon_1'$, in the sequel we consider any solution $\varepsilon$ to (7) such that $0 < \varepsilon < \varepsilon_1'$ (if such a solution exists).

By the assumption on $\log M\big(u, F, \|\cdot\|_{L^p(\mu)}\big)$ for $u = 3\varepsilon \leq \varepsilon_0$, and then using (7), we have, setting $r = 2(b-a)$,

$$c_0(3\varepsilon)^{-\alpha} \leq \log M\big(3\varepsilon, F, \|\cdot\|_{L^p(\mu)}\big) \leq c \operatorname{fat}_{\frac{\varepsilon}{32}}(G) \log^2 \left( \frac{r \operatorname{fat}_{\frac{\varepsilon}{32}}(G)}{\varepsilon} \right) \leq cP \log^2 \left( \frac{rP}{\varepsilon} \right) \ ,$$

where the last inequality is because $t \mapsto ct \log^2\left(\frac{rt}{\varepsilon}\right)$ is non-decreasing on $[\varepsilon/r, +\infty)$, with $\varepsilon/r \leq 1$, and $1 \leq \operatorname{fat}_{\frac{\varepsilon}{32}}(G) \leq \operatorname{Pdim}(G) = P$ (the lower bound of 1 follows from $c_0(3\varepsilon)^{-\alpha} > 0$).

Solving the inequation $c_0(3\varepsilon)^{-\alpha} \leq cP \log^2(rP/\varepsilon)$ for $\varepsilon$ (see Appendix B.3 for details), we get

$$\varepsilon \geq \min\big\{\varepsilon_1'', \ c_1 P^{-\frac{1}{\alpha}} \log^{-\frac{2}{\alpha}} P\big\} \ , \tag{9}$$

for some constants $\varepsilon_1'', c_1 > 0$ depending only on $p, c_0, b-a$ and $\alpha$. Setting $\varepsilon_1 = \min\{\varepsilon_1'', \varepsilon_1'\}$ and noting that $\varepsilon_1'$ only depends on $\varepsilon_0$ and $b-a$, we conclude the proof. $\qquad\square$

## 3 Approximation of Hölder balls by feed-forward neural networks

In this section, we apply Corollary 1 to establish nearly-tight lower bounds for the approximation of unit Hölder balls by feed-forward neural networks. Our main result is Proposition 3, which solves an open question by [DHP21].

Throughout the section, for any $s > 0$, we denote by $n$ and $\alpha$ the unique members of the decomposition $s = n + \alpha$ such that $n \in \mathbb{N}$ and $0 < \alpha \leq 1$.

For a set $\mathcal{X} \subset \mathbb{R}^d$, we follow [YZ20] and define the Hölder space $\mathcal{C}^{n,\alpha}(\mathcal{X})$ as the space of $n$ times continuously differentiable functions with finite norm

$$\|f\|_{\mathcal{C}^{n,\alpha}} = \max \left\{ \max_{\mathbf{n}:|\mathbf{n}| \leq n} \|D^{\mathbf{n}}f\|_\infty, \max_{\mathbf{n}:|\mathbf{n}|=n} \sup_{x \neq y} \frac{|D^{\mathbf{n}}f(x) - D^{\mathbf{n}}f(y)|}{\|x-y\|_2^\alpha} \right\},$$

where, for $\mathbf{n} = (n_1, \cdots, n_d) \in \mathbb{N}^d$, $D^{\mathbf{n}}f = \left(\frac{\partial}{\partial x_1}\right)^{n_1} \cdots \left(\frac{\partial}{\partial x_d}\right)^{n_d} f$ denotes the $|\mathbf{n}|$-order partial derivative of $f$. We denote

$$F_{s,d} = \{f \in \mathcal{C}^{n,\alpha}([0,1]^d) : \|f\|_{\mathcal{C}^{n,\alpha}} \leq 1\}.$$

Let $\lambda$ denote the Lebesgue measure over $[0,1]^d$. In this section, we provide nearly matching upper and lower bounds for the $L^p(\lambda)$ approximation error of elements of $F_{s,d}$ by feed-forward ReLU neural networks. The bounds are expressed in terms of the number of weights of the network.

## 3.1 Known bounds on the sup norm approximation error

[YZ20] gives matching (up to a certain constant) lower and upper bounds of the sup norm approximation error of the elements of $F_{s,d}$ by feed-forward ReLU neural networks.

**Proposition 2** ([YZ20]). *Let $d \in \mathbb{N}^*$, $s > 0$, $\gamma \in \left(\frac{s}{d}, \frac{2s}{d}\right]$. Consider $n \in \mathbb{N}$ and $\alpha \in (0,1]$ such that $s = n + \alpha$.*

*There exist positive constants $W_{\min}$ and $c_1$, depending only on $d$ and $n$, such that for any integer $W \geq W_{\min}$, there exists a feed-forward ReLU neural network architecture $\mathcal{A}$ with $L = O(W^{\gamma \frac{d}{s}-1})$ layers and $W$ weights such that*

$$\sup_{f \in F_{s,d}} \inf_{g \in H_{\mathcal{A}}} \|f-g\|_\infty \leq c_1 W^{-\gamma}. \tag{10}$$

*In the meantime, there exists a constant $c_2 > 0$ depending only on $d$ and $n$ such that, for any feed-forward neural network architecture $\mathcal{A}$ with $W$ weights and $L = o(W^{\gamma \frac{d}{s}-1}/\log W)$ layers and for the ReLU activation function,*

$$\sup_{f \in F_{s,d}} \inf_{g \in H_{\mathcal{A}}} \|f-g\|_\infty \geq c_2 W^{-\gamma}. \tag{11}$$

It is worth stressing that, for any probability measure $\mu$ on $[0,1]^d$, the upper bound (10) is automatically generalized to any smaller $L^p(\mu)$ norm, when $1 \leq p < +\infty$. However, the lower bound (11) does not immediately apply when $\|\cdot\|_\infty$ is replaced with $\|\cdot\|_{L^p(\mu)}$, $1 \leq p < +\infty$. The lower bound of the next subsection shows that, in this setting, approximation in $L^p(\lambda)$ norm is not easier than in sup norm, solving an open question of DeVore et al. [DHP21].

## 3.2 Nearly-matching lower bounds of the $L^p(\lambda)$ approximation error

We first state a lower bound on the packing number of $F_{s,d}$, which is rather classical though hard to find in this specific form (see [BS67] for the $L^\infty$ norm, or [ET96] for other Sobolev-type norms). For the sake of completeness, we give a proof of Lemma 2 in the supplement, Appendix D.1.

**Lemma 2.** *Let $s > 0$, $d \in \mathbb{N}^*$ and $1 \leq p < +\infty$. There exist constants $\varepsilon_0, c_0 > 0$ such that for any $0 < \varepsilon \leq \varepsilon_0$,*

$$\log M\left(\varepsilon, F_{s,d}, \|\cdot\|_{L^p(\lambda)}\right) \geq c_0 \varepsilon^{-\frac{d}{s}}. \tag{12}$$

Given Lemma 2, we can use Corollary 1 to establish the next proposition and obtain the lower bound on the $L^p(\lambda)$ approximation error.

**Proposition 3.** *Let $d \in \mathbb{N}^*$, $s > 0$, $\gamma \in \left(\frac{s}{d}, \frac{2s}{d}\right]$ and $1 \leq p < +\infty$. Consider $n \in \mathbb{N}$ and $\alpha \in (0,1]$ such that $s = n + \alpha$.*

*Let $\sigma : \mathbb{R} \to \mathbb{R}$ be a piecewise-affine function, and $c > 0$. Then, there exist $c_1 > 0$ and $W_{\min} \in \mathbb{N}^*$ (depending only on $s$, $d$, $p$, $\sigma$ and $c$) such that for any architecture $\mathcal{A}$ of depth $1 \leq L \leq cW^{\gamma \frac{d}{s}-1}$ with $W \geq W_{\min}$ weights, and for the activation $\sigma$, the set $H_{\mathcal{A}}$ (cf. Section 1) satisfies*

$$\sup_{f \in F_{s,d}} \inf_{g \in H_{\mathcal{A}}} \|f-g\|_{L^p(\lambda)} \geq c_1 W^{-\gamma} \log^{-\frac{3s}{d}}(W). \tag{13}$$

Note that the rate of the lower bound does not depend on $p$. Note also that, when the activation function is ReLU (which is piecewise-affine), we obtain a lower bound which matches the upper bound of the previous subsection up to logarithmic factors.

*Proof.* From Lemma 2, there exist $\varepsilon_0, c_0 > 0$ such that $\log M\left(\varepsilon, F_{s,d}, \|\cdot\|_{L^p(\lambda)}\right) \geq c_0 \varepsilon^{-\frac{d}{s}}$ for all $0 < \varepsilon \leq \varepsilon_0$. Combining with Corollary 1 and using $L \leq cW^{\gamma\frac{d}{s}-1}$ concludes the proof. $\qquad\square$

**Remark 1** (Comparison with existing proof strategies in sup norm.)**.** *We would like to highlight a key difference between the proof of Proposition 3 and the lower bound proof strategies of [Yar17, Yar18, YZ20, SYZ22] that are specific to the sup norm. Their overall argument is roughly the following: if $G$ can approximate any $f \in F$ in sup norm at accuracy $\varepsilon > 0$, since $F$ contains many "oscillating" functions with oscillation amplitude roughly $\varepsilon$, then so must be the case for $G$ (the sup norm is key here: **all** oscillations of any $f \in F$ are well approximated). Therefore, a small $\varepsilon$ implies a large* $\mathrm{VCdim}(G)$*, which by contrapositive enables to lower bound the approximation error* (1) *with a decreasing function of* $\mathrm{VCdim}(G)$*, and therefore as a function of $L$ and $W$. In contrast, in the proof of Theorem 1, the key probability result of Mendelson (Proposition 1) enables us to show that, even if the oscillations of any $f \in F$ are only well approximated **on average** (in $L^p(\mu)$ norm) by $G$, then* $\mathrm{Pdim}(G)$ *must be large when $\varepsilon$ is small. The conclusion is then the same: the approximation error in $L^p(\mu)$ norm can be lower bounded as a function of* $\mathrm{Pdim}(G)$*, and therefore in terms of $L$, $W$. This solves the question of DeVore et al. [DHP21] mentioned in the introduction, showing in particular that VC dimension theory can (surprisingly) be useful to prove $L^p$ approximation lower bounds.*

# 4 Approximation of monotonic functions by feed-forward neural networks

In this section, we consider the problem of approximating the set $\mathcal{M}^d$ of all non-decreasing functions from $[0,1]^d$ to $[0,1]$. These are functions $f : [0,1]^d \to [0,1]$ that are non-decreasing along any line parallel to an axis, i.e., such that, for all $x, y \in [0,1]^d$,

$$x_i \leq y_i, \ \forall i = 1, \dots, d \implies f(x) \leq f(y) .$$

Monotonic functions are an interesting case study for at least two reasons. First, they naturally appear in physics or engineering applications (consider for instance the braking distance of a vehicle as a function of variables such as the speed, the total load or the drag coefficient). Second, as will be shown in this section, because their sets of discontinuities can have "complex" shapes in dimension $d \geq 2$, monotonic functions provide a good example for which the approximation by feed-forward neural networks is hopeless in sup norm, but can be achieved in $L^p(\lambda)$ norm.

Next we focus on the approximation of $\mathcal{M}^d$ with Heaviside feed-forward neural networks. After proving an impossibility result for the sup norm, we show that the weaker goal of approximating $\mathcal{M}^d$ in $L^p(\lambda)$ norm is feasible, and derive nearly matching lower and upper bounds. Interestingly, the approximation rates depend on $p \geq 1$, which is in sharp contrast with the case of Hölder balls, that are not easier to approximate in $L^p(\lambda)$ norm than in sup norm (see Section 3).

## 4.1 Warmup: an impossibility result in sup norm

We start this section by showing that approximating monotonic functions of $d \geq 2$ variables in sup norm is impossible with Heaviside neural networks.

**Proposition 4.** *For any neural network architecture $\mathcal{A}$ with the Heaviside activation, the set $H_{\mathcal{A}}$ (cf. Section 1) satisfies*

$$\sup_{f \in \mathcal{M}^d} \inf_{g \in H_{\mathcal{A}}} \|f - g\|_\infty \geq \frac{1}{2}.$$

The proof of Proposition 4 is postponed to the supplement, Appendix E.2. We show a slightly stronger result, by exhibiting a single function $f \in \mathcal{M}^d$ such that the lower bound of $\frac{1}{2}$ holds simultaneously for all network architectures.

Next we study the approximation of $\mathcal{M}^d$ in $L^p(\lambda)$ norm.

## 4.2 Lower bound in $L^p(\lambda)$ norm

We start by proving a lower bound, as a direct consequence of Corollary 1 and a lower bound on the packing number due to [GW07].

**Proposition 5.** *Let $1 \leq p < +\infty$, $d \geq 1$, and let $\alpha = \max\{d, (d-1)p\}$. Let $\sigma : \mathbb{R} \to \mathbb{R}$ be a piecewise-polynomial function having maximal degree $\nu \in \mathbb{N}$. Then, there exist positive constants $c_1, c_2, c_3, W_{\min}$ (depending only on $d$, $p$, and $\sigma$) such that for any architecture $\mathcal{A}$ of depth $L \geq 1$ with $W \geq W_{\min}$ weights, and for the activation $\sigma$, the set $H_{\mathcal{A}}$ (cf. Section 1) satisfies*

$$\sup_{f \in \mathcal{M}^d} \inf_{g \in H_{\mathcal{A}}} \|f - g\|_{L^p(\lambda)} \geq \begin{cases} c_1 W^{-\frac{2}{\alpha}} \log^{-\frac{2}{\alpha}}(W) & \text{if } \nu \geq 2, \\ c_2 (LW)^{-\frac{1}{\alpha}} \log^{-\frac{3}{\alpha}}(W) & \text{if } \nu = 1, \\ c_3 W^{-\frac{1}{\alpha}} \log^{-\frac{3}{\alpha}}(W) & \text{if } \nu = 0. \end{cases} \qquad (14)$$

Note that, contrary to the case of Hölder balls (Section 3), the rate of the lower bound depends on $p$ through $\alpha = \max\{d, (d-1)p\}$.

*Proof.* From [GW07], there exist constants $\varepsilon_0, c_0 > 0$ such that for $\varepsilon \leq \varepsilon_0$, $\log M\left(\varepsilon, \mathcal{M}_d, \|\cdot\|_{L^p(\lambda)}\right) \geq c_0 \varepsilon^{-\alpha}$. Using Corollary 1, we obtain the result. □

## 4.3 Nearly-matching upper bound in $L^p(\lambda)$ norm

To the best of our knowledge, there does not exist any upper-bound of the $L^p(\lambda)$ approximation error of $\mathcal{M}^d$ with feed-forward neural networks. Checking that all the lower-bounds of Proposition 5 are tight is out of the scope of this paper and we leave it for future research[3]. However, we establish in the next proposition upper-bounds of the $L^p(\lambda)$ approximation error of $\mathcal{M}^d$ with feed-forward neural networks with the Heaviside activation function. This shows that, for the $L^p(\lambda)$ approximation error, the lower-bound obtained in (14), for $\nu = 0$, is tight up to logarithmic factors. The next proposition follows by reinterpreting a metric entropy upper bound of [GW07] in terms of Heaviside neural networks. The proof is postponed to Appendix E.1 in the supplement.

**Proposition 6.** *Let $1 \leq p < +\infty$, $d \in \mathbb{N} \setminus \{0, 1\}$ and let $\alpha = \max\{d, (d-1)p\}$. There exist positive constants $W_{\min}$ and $c$, depending only on $d$ and $p$, such that for any integer $W \geq W_{\min}$, there exists a feed-forward architecture $\mathcal{A}$ with two hidden layers, $W$ weights and the Heaviside activation function such that the set $H_{\mathcal{A}}$ satisfies*

$$\sup_{f \in \mathcal{M}^d} \inf_{g \in H_{\mathcal{A}}} \|f - g\|_{L^p(\lambda)} \leq \begin{cases} c W^{-\frac{1}{\alpha}} & \text{if } p(d-1) \neq d, \\ c W^{-\frac{1}{d}} \log(W) & \text{if } p(d-1) = d. \end{cases} \qquad (15)$$

## 5 Conclusion and other possible applications

We proved a general lower bound on the approximation error of $F$ by $G$ in $L^p(\mu)$ norm (Theorem 1), in terms of generic properties of $F$ and $G$ (packing number of $F$, range of $F$, fat-shattering dimension of $G$). The proof relies on VC dimension theory as in the sup norm case, but uses an additional key probabilistic argument due to Mendelson ([Men02], see Proposition 1), solving a question raised by DeVore et al. [DHP21].

In Sections 3 and 4 we detailed two applications, where Corollary 1 yields nearly optimal approximation lower bounds in $L^p$ norm, and which correspond to two examples where the approximation rate may depend or not depend on $p$.

Theorem 1 and Corollary 1 can be used to derive approximation lower bounds for many other cases. Corollary 1 only requires a (tight) lower bound on the packing number of $F$, for which approximation theory provides several examples. For instance, for the *Barron space* introduced in [Bar93], Petersen and Voigtlaender [PV21] showed a tight lower bound on the log packing number in $L^p(\lambda, [0,1]^d)$ norm, of order $\varepsilon^{-2d/(d+2)}$. Applying Corollary 1, this yields an approximation lower bound of

---

[3]Obtaining an upper-bound for ReLU networks seems challenging. For example, the bit extraction technique used in [Yar18] to find a sharp upper bound heavily relies on the local smoothness assumption of the function to approximate, which is not satisfied in general for monotonic functions.

$(LW)^{-\left(\frac{1}{2}+\frac{1}{d}\right)}\log^{-3\left(\frac{1}{2}+\frac{1}{d}\right)}(W)$ for ReLU networks (see Appendix F in the supplement for details). Other examples of sets $F$ for which tight lower bounds on the packing number (or metric entropy) are available include: multivariate cumulative distribution functions [BGL07], multivariate convex functions [GS13], and functions with other shape constraints [GJ14].

Piecewise-polynomial activation functions are not essential for the current derivation. Indeed, Theorem 1 can also be applied to the case where $G$ corresponds to a neural network with other activation functions such as the sigmoid. In the sigmoid case, the pseudo-dimension is known to be at most of the order of $W^4$ (see [KM97, AB99]), which we can use to derive an approximation lower bound similar to that of Corollary 1, with a smaller right-hand side for large $W$. However, to the best of our knowledge, it is not known whether the $\mathcal{O}(W^4)$ VC bound is tight (only a lower bound of the order of $W^2$ is known), so the resulting approximation lower bound could be loose. We leave this interesting question for future work.

Theorem 1 can also be applied to other approximating sets $G$, beyond classical feed-forward neural networks, as soon as a (tight) upper bound on the fat-shattering dimension of $G$ is available. For example, upper bounds were derived by [WS22] on the VC dimension of partially quantized networks, while [Bel18] derived bounds on the fat-shattering dimension of some RKHS. Investigating such applications and whether the obtained approximation lower bounds are rate-optimal is a natural research direction for the future.

## Acknowledgements

The authors would like to thank Keridwen Codet for contributing to the results of Sections 4.1 and 4.3.

This work has benefited from the AI Interdisciplinary Institute ANITI, which is funded by the French "Investing for the Future – PIA3" program under the Grant agreement ANR-19-P3IA-0004. The authors gratefully acknowledge the support of IRT Saint Exupéry and the DEEL project.[4]

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
