# A general approximation lower bound in $L^p$ norm, with applications to feed-forward neural networks
## Supplementary Material

This is the appendix for "A general approximation lower bound in $L^p$ norm, with applications to feed-forward neural networks".

## A    Feed-forward neural networks: formal definition

In all this paper, we use the following classical graph-theoretic definitions for feed-forward neural networks given, e.g., in [BHLM19] (with slightly different terms and notation).

A *feed-forward neural network architecture* $\mathcal{A}$ of depth $L \geq 1$ is a directed acyclic graph $(V, E)$ with $d \geq 1$ nodes with in-degree 0 (also called the *input neurons*), a single node with out-degree 0 (also called the *output neuron*), and such that the longest path in the graph has length $L$.

We define layers $\ell = 0, 1, \ldots, L$ recursively as follows:

- layer 0 is the set $V_0$ of all input neurons; we assume that $V_0 = \{1, \ldots, d\}$ without loss of generality.

- for any $\ell = 1, \ldots, L$, layer $\ell$ is the set $V_\ell$ of all nodes that have one or several predecessors[5] in layer $\ell - 1$, possibly other predecessors in layers $0, 1, \ldots, \ell - 2$, but no other predecessors.

Layer $L$ consists of a single node: the output neuron. Layers $1, \ldots, L - 1$ are called the *hidden layers* (if $L \geq 2$). Note that skip connections are allowed, i.e., there can be connections between non-consecutive layers.

Given a feed-forward neural network architecture $\mathcal{A}$ of depth $L \geq 1$, we associate real numbers $w_e \in \mathbb{R}$ to all edges $e \in E$ and $w_v \in \mathbb{R}$ to all nodes $v \in V_1 \cup \ldots \cup V_L$. These real numbers are called *weights* (they correspond to linear coefficients and biases) and are concatenated in a *weight vector* $\mathbf{w} \in \mathbb{R}^W$, where $W = \mathrm{Card}(E) + \sum_{\ell=1}^{L} \mathrm{Card}(V_\ell)$ is the total number of weights.

Given $\mathcal{A}$, an associated weight vector $\mathbf{w} \in \mathbb{R}^W$, and a function $\sigma : \mathbb{R} \to \mathbb{R}$ (called *activation function*), the network represents the function $g_{\mathbf{w}} : \mathbb{R}^d \to \mathbb{R}$ defined recursively as follows. We write $P_v \subset V$ for the set of all predecessors of any node $v \in V$, and $w_{u \to v}$ for the weight on the edge from $u$ to $v$. The recursion from layer $\ell = 0$ to layer $\ell = L$ reads: given $x = (x_1, \ldots, x_d) \in \mathbb{R}^d$,

- each input neuron $v \in \{1, \ldots, d\}$ outputs the value $y_v := x_v$;

- for any $\ell = 1, \ldots, L - 1$, each neuron $v \in V_\ell$ outputs $y_v := \sigma\left(\sum_{u \in P_v} w_{u \to v} y_u + w_v\right)$;

- the unique output neuron $v \in V_L$ outputs $g_{\mathbf{w}}(x) := \sum_{u \in P_v} w_{u \to v} y_u + w_v$.

Finally, we define $H_{\mathcal{A}} := \{g_{\mathbf{w}} : \mathbf{w} \in \mathbb{R}^W\}$ to be the set of all functions that can be represented by tuning all the weights assigned to the network (the dependency on the activation function $\sigma$ is not written explicitly).

## B    Main results: technical details

We provide technical details that were missing to establish Proposition 1, Theorem 1 and Corollary 1.

---

[5]A node $u \in V$ is a predecessor of another node $v \in V$ if there is a directed edge from $u$ to $v$.

## B.1 Proof of Proposition 1

Proposition 1 is a direct extension of [Men02, Corollary 3.12] to any range $[a, b]$. We first recall this result but in slightly different terms (see the comments afterwards).

**Proposition 7** (Corollary 3.12 in [Men02], "almost equivalent" statement). *Let $G$ be a set of measurable functions from a measurable space $\mathcal{X}$ to $[0, 1]$, such that $\mathrm{fat}_\gamma(G) < +\infty$ for all $\gamma > 0$. Then, for every $1 \leq p < +\infty$, there is some constant $c_p > 0$ depending only on $p$ such that, for every probability measure $\mu$ on $\mathcal{X}$ and every $\varepsilon > 0$,*

$$\log M\left(\varepsilon, G, \|\cdot\|_{L^p(\mu)}\right) \leq c_p \, \mathrm{fat}_{\frac{\varepsilon}{32}}(G) \log^2\left(\frac{2 \, \mathrm{fat}_{\frac{\varepsilon}{32}}(G)}{\varepsilon}\right).$$

To be precise, [Men02, Corollary 3.12] was stated a little differently. Instead of the assumption on $\mathrm{fat}_\gamma(G)$, there were two conditions on $G$: (i) $G$ satisfies a weak measurability assumption such as the "image admissible Suslin" property, and (ii) $G$ is a uniform Glivenko-Cantelli class. Fortunately, note that assumption (i) could easily be checked in special cases such as the setting of Corollary 1, and that assumption (ii) is equivalent to $\mathrm{fat}_\gamma(G) < +\infty$ for all $\gamma > 0$ when (i) holds and when $G$ only consists of $[0, 1]$-valued functions (see [ABDCBH97], Theorem 2.5). The two statements are thus "almost equivalent". However, we stress that (i) and (ii) are not necessary (assuming $\mathrm{fat}_\gamma(G) < +\infty$ for all $\gamma > 0$). To see why, it suffices to adapt the proof of [Men02, Corollary 3.12] as follows: instead of starting from an $\varepsilon$-packing of $G$ in empirical $L^p(\mu_n)$ norm and showing that it is also an $\varepsilon'$-covering of $G$ in $L^p(\mu)$ norm, with $\varepsilon' > \varepsilon$, we can start from an $\varepsilon$-packing of $G$ in $L^p(\mu)$ norm and show that it is also an $\varepsilon$-packing of $G$ in empirical $L^p(\mu_n)$ norm for some large integer $n$ (with positive probability). This last statement directly follows from the Hoeffding inequality: no uniform law of large numbers is required, since we only need to compare empirical averages to their expectations for a finite number of bounded functions.[6]

We now explain how to derive Proposition 1 (with an arbitrary range $[a, b]$) as a straightforward consequence of Proposition 7.

*Proof (of Proposition 1).* In order to apply Proposition 7, we reduce the problem from $[a, b]$ to $[0, 1]$ by translating and rescaling every function in $G$. For $g \in G$, we define $\tilde{g} : \mathcal{X} \to [0, 1]$ by $\tilde{g}(x) = \frac{g(x) - a}{b - a}$, and we set

$$\tilde{G} = \{\tilde{g} \ : \ g \in G\} \ .$$

Note that every $\tilde{g} \in \tilde{G}$ is indeed $[0, 1]$-valued.

We now note that translation does not affect packing numbers nor the fat-shattering dimension, while rescaling only changes the scale $\varepsilon$ by a factor of $b - a$. More precisely, we have the following two properties:

**Property 1:** For all $u > 0$, $\mathrm{fat}_{\frac{u}{b-a}}(\tilde{G}) = \mathrm{fat}_u(G)$.
**Property 2:** For all $u > 0$, $M\left(\frac{u}{b-a}, \tilde{G}, \|\cdot\|_{L^p(\mu)}\right) = M\left(u, G, \|\cdot\|_{L^p(\mu)}\right)$.

Before proving the two properties (see below), we first conclude the proof of Proposition 1. By Property 1, $\mathrm{fat}_\gamma(\tilde{G}) = \mathrm{fat}_{\gamma(b-a)}(G)$, which by assumption is finite for all $\gamma > 0$. Since every $\tilde{g} \in \tilde{G}$ is $[0, 1]$-valued, we can thus apply Proposition 7. Using it with $\tilde{\varepsilon} = \varepsilon/(b - a)$, we get

$$\log M\left(\tilde{\varepsilon}, \tilde{G}, \|\cdot\|_{L^p(\mu)}\right) \leq c_p \, \mathrm{fat}_{\frac{\tilde{\varepsilon}}{32}}(\tilde{G}) \log^2\left(\frac{2 \, \mathrm{fat}_{\frac{\tilde{\varepsilon}}{32}}(\tilde{G})}{\tilde{\varepsilon}}\right) \ .$$

Combining with the two equalities in Properties 1 and 2, we obtain

$$\log M\left(\varepsilon, G, \|\cdot\|_{L^p(\mu)}\right) \leq c_p \, \mathrm{fat}_{\frac{\varepsilon}{32}}(G) \log^2\left(\frac{2(b - a) \, \mathrm{fat}_{\frac{\varepsilon}{32}}(G)}{\varepsilon}\right) \ ,$$

which concludes the proof of Proposition 1.

We now prove the two properties.

---

[6]In passing, all occurrences of $\mathrm{fat}_{\frac{\varepsilon}{32}}(G)$ could be replaced with $\mathrm{fat}_{\frac{\varepsilon}{8}}(G)$.

**Proof of Property 1.** We first show that $\text{fat}_{\frac{u}{b-a}}(\tilde{G}) \geq \text{fat}_u(G)$. To that end, let $S = \{x_1, \ldots, x_m\}$ and $r : S \to \mathbb{R}$ be such that for any $E \subset S$, there exists $g \in G$ such that $g(x) \geq r(x) + u$ if $x \in E$ and $g(x) \leq r(x) - u$ otherwise. Setting $\tilde{r}(x) = \frac{r(x)-a}{b-a}$, we can see that $\tilde{g}(x) \geq \tilde{r}(x) + \frac{u}{b-a}$ if $x \in E$ and $\tilde{g}(x) \leq \tilde{r}(x) - \frac{u}{b-a}$ otherwise, which proves $\text{fat}_{\frac{u}{b-a}}(\tilde{G}) \geq \text{fat}_u(G)$. The reverse inequality is proved similarly.

**Proof of Property 2.** Let $\{g_1, \ldots, g_m\}$ be a $u$-packing of $G$ in $L^p(\mu)$ norm. This means that $\|g_i - g_j\|_{L^p(\mu)} > u$ and therefore $\|\tilde{g}_i - \tilde{g}_j\|_{L^p(\mu)} > \frac{u}{b-a}$ for all $i \neq j \in \{1, \ldots, m\}$, so that $\{\tilde{g}_1, \ldots, \tilde{g}_m\} \subset \tilde{G}$ is a $\frac{u}{b-a}$-packing of $\tilde{G}$. This proves $M\left(\frac{u}{b-a}, \tilde{G}, \|\cdot\|_{L^p(\mu)}\right) \geq M\left(u, G, \|\cdot\|_{L^p(\mu)}\right)$. The reverse inequality is proved similarly. $\qquad\square$

## B.2 Clipping can only help

The next two lemmas indicate that clipping (truncature) to a known range can only help. These are key to apply Proposition 1 in our setting. In the sequel, for a set $G$ of functions from a set $\mathcal{X}$ to $\mathbb{R}$, and for $a < b$ in $\mathbb{R}$, we denote by $G_{[a,b]}$ the set of all functions in $G$ whose values are truncated (clipped) to the segment $[a, b]$, that is, $G_{[a,b]} = \{\tilde{g} : g \in G\}$, where $\tilde{g} : \mathcal{X} \to \mathbb{R}$ is given by

$$\forall x \in \mathcal{X}, \quad \tilde{g}(x) = \min(\max(a, g(x)), b) .$$

**Lemma 3.** *Let $G$ be a set of functions defined on a set $\mathcal{X}$, and with values in $\mathbb{R}$. Let $G_{[a,b]}$ be defined as above. Then, for any $\gamma > 0$,*

$$\text{fat}_\gamma(G) \geq \text{fat}_\gamma(G_{[a,b]}) .$$

*Proof.* Let $\gamma > 0$. The case when $\text{fat}_\gamma(G_{[a,b]}) = 0$ is straightforward. We thus assume that $\text{fat}_\gamma(G_{[a,b]}) \geq 1$. To prove the result, we show that any subset $A$ of $X$ that is $\gamma$-*shattered* by $G_{[a,b]}$ is also $\gamma$-*shattered* by $G$. Let us consider such a subset $A = \{x^1, \ldots, x^N\} \subset X$, with cardinality $N \geq 1$. Hence, there exists $\{r_1, \ldots, r_N\} \subset \mathbb{R}$ such that for any $E \subset A$, there exists $\tilde{g} \in G_{[a,b]}$ such that $\tilde{g}(x_i) - r_i \geq \gamma$ if $x_i \in E$ and $\tilde{g}(x_i) - r_i \leq -\gamma$ otherwise. Note that this must imply that $r_i \in ]a, b[$ for all $i = 1, \ldots, N$ (indeed, by choosing $E$ such that $x_i \in E$ or not, we have either $r_i + \gamma \leq \tilde{g}(x_i) \leq b$ or $r_i - \gamma \geq \tilde{g}(x_i) \geq a$). Now fix $i \in \{1, \ldots, N\}$ and let us assume $\tilde{g}(x_i) - r_i \geq \gamma$ (by symmetry, the reversed case $\tilde{g}(x_i) - r_i \leq -\gamma$ is treated the same way). Because $r_i > a$, this implies that $\tilde{g}(x_i) > a$ and thus $g(x_i) \geq \tilde{g}(x_i)$ (by definition of $\tilde{g}$), which entails $g(x_i) - r_i \geq \gamma$. It follows that if $G_{[a,b]}$ $\gamma$-*shatters* $A$, then $G$ also $\gamma$-*shatters* $A$, and the result follows. $\qquad\square$

The following lemma formalizes the well-known idea that it is easier to approach a function with values in a finite range by a function with values in the same range.

**Lemma 4.** *Let $G$ be a set of measurable functions from a measurable space $\mathcal{X}$ to $\mathbb{R}$, and let $G_{[a,b]}$ be defined as above. Assume $F$ is a set of measurable functions from $\mathcal{X}$ to $[a, b]$. Then, for any probability measure $\mu$ on $\mathcal{X}$,*

$$\sup_{f \in F} \inf_{g \in G} \|f - g\|_{L^p(\mu)} \geq \sup_{f \in F} \inf_{\tilde{g} \in G_{[a,b]}} \|f - \tilde{g}\|_{L^p(\mu)} .$$

*Proof.* To prove the above result, it is enough to show that for any $f \in F$ and $g \in G$, the function $\tilde{g}$ is pointwise at least as close to $f$ as $g$ is, which for all $f \in F$ yields $\inf_{g \in G} \|f - g\|_{L^p(\mu)} \geq \inf_{\tilde{g} \in G_{[a,b]}} \|f - \tilde{g}\|_{L^p(\mu)}$. By definition of $G_{[a,b]}$, for any $x \in \mathcal{X}$, if $g(x) \in [a, b]$, then $|f(x) - g(x)| = |f(x) - \tilde{g}(x)|$. And if $g(x) \notin [a, b]$, then $|f(x) - \tilde{g}(x)| < |f(x) - g(x)|$ since $f(x) \in [a, b]$. It follows that the discrepancy $|f - \tilde{g}|$ is everywhere bounded by $|f - g|$, and the result follows. $\qquad\square$

## B.3 Missing details in the proof of Theorem 1

We provide all details that were missing to derive (9), which is a direct consequence of Lemma 5 below. We follow the convention $aP^{-\frac{1}{\alpha}} \log^{-\frac{2}{\alpha}}(P) = +\infty$ when $P = 1$.

**Lemma 5.** *Let $P \in \mathbb{N}^*$ and $c, \alpha, r > 0$. There exist constants $a, \varepsilon_1'' > 0$ depending only on $c$, $\alpha$ and $r$ such that, for all $\varepsilon \in (0, r)$ satisfying*

$$\varepsilon^{-\alpha} \leq cP \log^2\left(\frac{rP}{\varepsilon}\right) , \tag{16}$$

*we have*

$$\varepsilon \geq \min\left(\varepsilon_1'', aP^{-\frac{1}{\alpha}}\log^{-\frac{2}{\alpha}}(P)\right).$$

*Proof.* Assume $\varepsilon \in (0, r)$ is such that (16) holds. To show the result, we study the function $f : (1/r, +\infty) \to \mathbb{R}$ defined for all $x > 1/r$ by

$$f(x) = \frac{x^\alpha}{\log^2(rPx)} \ .$$

Note that (16) implies that $f(1/\varepsilon) \leq cP$. For all $P \geq 2$, we set

$$\varepsilon_P = P^{-\frac{1}{\alpha}}\log^{-\frac{2}{\alpha}}(P) \ . \tag{17}$$

Let $P_1 \geq 2$ be such that $P_1^{\frac{1}{\alpha}}\log^{\frac{2}{\alpha}}(P_1) \geq \frac{\exp(\frac{2}{\alpha})}{r}$. For all $P \geq P_1$, we have $\frac{1}{\varepsilon_P} \geq \frac{\exp(\frac{2}{\alpha})}{r} > 1/r$ and

$$f\left(\frac{1}{\varepsilon_P}\right) = \frac{P\log^2(P)}{\log^2\left(rP^{1+\frac{1}{\alpha}}\log^{\frac{2}{\alpha}}(P)\right)} \ .$$

Since

$$\lim_{Q \to +\infty} \frac{\log^2(Q)}{\log^2\left(rQ^{1+\frac{1}{\alpha}}\log^{\frac{2}{\alpha}}(Q)\right)} = \frac{1}{(1+\frac{1}{\alpha})^2} =: c_1 \ ,$$

there exists $P_2$ such that for all $Q \geq P_2$, we have $\frac{\log^2(Q)}{\log^2\left(rQ^{1+\frac{1}{\alpha}}\log^{\frac{2}{\alpha}}(Q)\right)} \geq \frac{c_1}{2}$ .

Below we distinguish the cases $P \geq \max(P_1, P_2)$ and $P < \max(P_1, P_2)$.

**1st case:** $P \geq \max(P_1, P_2)$.
We have $f\left(\frac{1}{\varepsilon_P}\right) \geq \frac{c_1 P}{2}$ and $P \geq \frac{1}{c}f\left(\frac{1}{\varepsilon}\right)$ (by (16)), so that $f\left(\frac{1}{\varepsilon_P}\right) \geq \frac{c_1}{2c}f\left(\frac{1}{\varepsilon}\right)$. We now use Lemma 6 below with $b = \frac{c_1}{2c}$: setting $a := (b/2)^{1/\alpha} = (c_1/(4c))^{1/\alpha}$, there exists $x_1 > \max\{\frac{1}{r}, \frac{1}{ar}\}$ depending only on $r, b, \alpha$ such that $bf(x) \geq f(ax)$ for all $x \geq x_1$.

Therefore, if $\varepsilon < \frac{1}{x_1} =: \varepsilon_1$, then $\frac{c_1}{2c}f\left(\frac{1}{\varepsilon}\right) \geq f\left(\frac{a}{\varepsilon}\right)$. Therefore $f\left(\frac{1}{\varepsilon_P}\right) \geq f\left(\frac{a}{\varepsilon}\right)$.

Recall from (17) and $P \geq P_1$ that $\frac{1}{\varepsilon_P} \geq \frac{\exp(\frac{2}{\alpha})}{r}$. If $\varepsilon < \frac{ar}{\exp(\frac{2}{\alpha})} =: \varepsilon_2$, then we also have $\frac{a}{\varepsilon} \geq \frac{\exp(\frac{2}{\alpha})}{r}$. Therefore, using Lemma 6 again, $f\left(\frac{1}{\varepsilon_P}\right) \geq f\left(\frac{a}{\varepsilon}\right)$ implies that $\frac{1}{\varepsilon_P} \geq \frac{a}{\varepsilon}$, that is,

$$\varepsilon \geq a\,\varepsilon_P \ .$$

Summarizing, when $\varepsilon \in (0, r)$ satisfies (16) and when $P \geq \max(P_1, P_2)$, either $\varepsilon \geq \varepsilon_1$ or $\varepsilon \geq \varepsilon_2$ or $\varepsilon \geq a\,\varepsilon_P$. Put differently,

$$\varepsilon \geq \min(\varepsilon_1, \varepsilon_2, a\,\varepsilon_P) \ . \tag{18}$$

**2nd case:** $P < \max(P_1, P_2) =: P_3$.
Using (16) and the fact that $t \mapsto ct\log^2\left(\frac{rt}{\varepsilon}\right)$ is non-decreasing on $[\varepsilon/r, +\infty)$, together with $\varepsilon/r \leq 1 \leq P \leq P_3$ yields $\varepsilon^{-\alpha} \leq cP_3\log^2(rP_3/\varepsilon)$. This entails that, for some $\varepsilon_3 > 0$ depending only on $\alpha, c, P_3, r$,

$$\varepsilon \geq \varepsilon_3 \ . \tag{19}$$

**Conclusion:** combining the two cases, when $\varepsilon \in (0, r)$ satisfies (16), whatever $P \in \mathbb{N}^*$, we have (18) or (19). Setting $\varepsilon_1'' = \min(\varepsilon_1, \varepsilon_2, \varepsilon_3)$, we obtain

$$\varepsilon \geq \min\left(\varepsilon_1'', a\,P^{-\frac{1}{\alpha}}\log^{-\frac{2}{\alpha}}(P)\right).$$

(Note that this is also true in the case $P = 1$, by the convention $aP^{-\frac{1}{\alpha}}\log^{-\frac{2}{\alpha}}(P) = +\infty$.) Since $\varepsilon_1, \varepsilon_2, \varepsilon_3$ and $a$ only depend on $c, \alpha, r$, this concludes the proof. $\qquad\square$

**Lemma 6.** *Let $\alpha, r > 0$ and $P \in \mathbb{N}^*$. We define $f(x) = \frac{x^\alpha}{\log^2(rPx)}$ for all $x > 1/r$. Then:*

*i)* $f$ *is increasing on* $I := \left[\frac{\exp(\frac{2}{\alpha})}{r}, +\infty\right)$ *and* $\lim_{x \to +\infty} f(x) = +\infty$.

*ii)* *for all* $b > 0$, *setting* $a := (b/2)^{1/\alpha}$, *there exists* $x_1 > \max\{\frac{1}{r}, \frac{1}{ar}\}$ *depending only on* $r, b, \alpha$ *such that,*

$$\forall x \geq x_1 , \qquad bf(x) \geq f(ax) .$$

*Proof.* **Proof of i):** The fact that $\lim_{x \to +\infty} f(x) = +\infty$ is because $\alpha > 0$. To see why $f$ is increasing on $I$, note that

$$f'(x) = \frac{\alpha x^{\alpha-1} \log^2(rPx) - x^\alpha 2 \log(rPx)\frac{1}{x}}{\log^4(rPx)} = \frac{x^{\alpha-1} \log(rPx)(\alpha \log(rPx) - 2)}{\log^4(rPx)} ,$$

so that $f'(x) > 0$ for all $x > \frac{\exp(\frac{2}{\alpha})}{rP}$, and in particular for all $x > \frac{\exp(\frac{2}{\alpha})}{r}$ (since $P \geq 1$). This proves that $f$ is increasing on $I$.

**Proof of ii):** Let $b > 0$ and set $a := (b/2)^{1/\alpha}$. Let $x_1 > \max\{\frac{1}{r}, \frac{1}{ar}\}$ depending only on $r, b, \alpha$ such that, for all $u \geq x_1$,

$$\frac{\log^2(ru)}{\log^2(rau)} \leq 2 .$$

(Such an $x_1$ exists since the ratio converges to 1 as $u \to +\infty$, and we can choose $x_1$ as a function of $r, a$ only.) Now, for all $x \geq x_1$, using the above inequality with $u = Px \geq x$ (since $P \geq 1$), we get

$$\frac{f(ax)}{f(x)} = a^\alpha \frac{\log^2(rPx)}{\log^2(rPax)} \leq 2a^\alpha = b ,$$

where the last equality is because $a := (b/2)^{1/\alpha}$. This proves that $bf(x) \geq f(ax)$ for all $x \geq x_1$. $\quad\square$

## B.4 Proof of Corollary 1

We first recall some definitions and two key bounds on the VC-dimension of piecewise-polynomial feed-forward neural networks, proved by [GJ95] and [BHLM19].

For a set $F$ of functions from $\mathcal{X}$ to $\{-1, 1\}$, we say that a set $S = \{x_1 \dots, x_N\} \subset \mathcal{X}$ is *shattered* by $F$ if for any $E \subset S$, there exists $f \in F$ satisfying for all $i = 1, \dots, N$, $f(x_i) = 1$ if $x_i \in E$, and $f(x_i) = -1$ if $x_i \notin E$. The VC-dimension of $F$, denoted by $\mathrm{VCdim}(F)$, is defined as the largest number $N \geq 1$ such that there exists $S \subset \mathcal{X}$ of cardinality $N$ which is shattered by $F$ (by convention, $\mathrm{VCdim}(F) = 0$ if no such set $S$ exists, while $\mathrm{VCdim}(F) = +\infty$ if there exist sets $S$ of unbounded cardinality $N$).

Let $\mathcal{B}$ be any feed-forward neural network architecture of depth $L \geq 1$ with $W \geq 1$ weights, $d \geq 1$ input neurons, and $U \geq 1$ hidden or output neurons. Let $\sigma : \mathbb{R} \to \mathbb{R}$ be any piecewise-polynomial activation function on $K \geq 2$ pieces, with maximal degree $\nu \in \mathbb{N}$. Denote by $\mathrm{sgn}(H_\mathcal{B}) = \{\mathrm{sgn}(g_\mathbf{w}) : \mathbf{w} \in \mathbb{R}^W\}$ the set of all classifiers obtained by looking at the sign of the network's output, that is, the classifiers defined by $\mathrm{sgn}(g_\mathbf{w})(x) = \mathbb{1}_{\{g_\mathbf{w}(x) > 0\}}$ for all $x \in \mathbb{R}^d$.

Goldberg and Jerrum [GJ95] showed that, for some constant $c_1' > 0$ depending only on $d$, $\nu$ and $K$, the VC-dimension of $\mathrm{sgn}(H_\mathcal{B})$ is bounded as follows (see also Theorem 8.7 in [AB99]):

$$\mathrm{VCdim}(\mathrm{sgn}(H_\mathcal{B})) \leq c_1' W^2 . \tag{20}$$

This bound was refined for piecewise-affine activation functions. Namely, Bartlett et al. [BHLM19, Theorem 7] proved that, if $U \geq 3$, then, for some $R \leq U + U(L-1)\nu^{L-1}$,

$$\mathrm{VCdim}(\mathrm{sgn}(H_\mathcal{B})) \leq L + \bar{L}W \log_2\left(4e(K-1)R\log_2\big(2e(K-1)R\big)\right) ,$$

where $\bar{L} = 1$ if $\nu = 0$, and $\bar{L} \leq L$ otherwise. Therefore, for some constants $W'_{\min} \geq 1$ and $c_2', c_3' > 0$ depending only on $d$ and $K$, we have, for all $W \geq W'_{\min}$ (which in particular implies $U \geq 3$),

$$\mathrm{VCdim}(\mathrm{sgn}(H_\mathcal{B})) \leq \begin{cases} c_2' LW \log(W) & \text{if } \nu = 1, \\ c_3' W \log(W) & \text{if } \nu = 0. \end{cases} \tag{21}$$

We are now ready to prove Corollary 1 from Theorem 1.

*Proof (of Corollary 1).* In order to apply Theorem 1, we first bound $P := \mathrm{Pdim}(H_{\mathcal{A}})$ from above. The bounds (20) and (21) were on the VC-dimension of $\mathrm{sgn}(H_{\mathcal{B}})$, for any feed-forward neural network architecture $\mathcal{B}$, while we need a bound on the pseudo-dimension. However, by a well-known trick (e.g., Theorem 14.1 in [AB99]), the pseudo-dimension of $H_{\mathcal{A}}$ is upper bounded by the VC-dimension of (the sign of) an augmented network architecture of depth $L$, with $d + 1$ input neurons and $W + 1$ weights.[7] Therefore, replacing $(d, W)$ with $(d + 1, W + 1)$ in (20) and (21), we get that, for some constants $\tilde{W}_{\min} \geq 1$ and $\tilde{c}_1, \tilde{c}_2, \tilde{c}_3 > 0$ depending only on $d, \nu$ and $K$, for all $W \geq \tilde{W}_{\min}$,

$$P \leq \begin{cases} \tilde{c}_1 W^2 & \text{if } \nu \geq 2 \,, \\ \tilde{c}_2 LW \log(W) & \text{if } \nu = 1 \,, \\ \tilde{c}_3 W \log(W) & \text{if } \nu = 0 \,. \end{cases} \tag{22}$$

Now, by Theorem 1, we have, for some constants $c_1, \varepsilon_1 > 0$ depending only on $b - a, p, c_0, \varepsilon_0, \alpha$,

$$\sup_{f \in F} \inf_{g \in H_{\mathcal{A}}} \|f - g\|_{L^p(\mu)} \geq \min\left\{\varepsilon_1, \; c_1 P^{-\frac{1}{\alpha}} \log^{-\frac{2}{\alpha}}(P)\right\} \,. \tag{23}$$

Noting that $P \mapsto \min\left\{\varepsilon_1, \; c_1 P^{-\frac{1}{\alpha}} \log^{-\frac{2}{\alpha}}(P)\right\}$ is non-increasing and plugging (22) into (23), we get, for $W \geq W_{\min}$,

$$\sup_{f \in F} \inf_{g \in H_{\mathcal{A}}} \|f - g\|_{L^p(\mu)} \geq \min\left\{\varepsilon_1, \; \begin{pmatrix} c_4 W^{-\frac{2}{\alpha}} \log^{-\frac{2}{\alpha}}(W^2) & \text{if } \nu \geq 2 \\ c_5 (LW \log(W))^{-\frac{1}{\alpha}} \log^{-\frac{2}{\alpha}}(LW \log(W)) & \text{if } \nu = 1 \\ c_6 (W \log(W))^{-\frac{1}{\alpha}} \log^{-\frac{2}{\alpha}}(W \log(W)) & \text{if } \nu = 0 \end{pmatrix} \right\}$$

for some constants $W_{\min} \geq 1$ and $c_4, c_5, c_6 > 0$ depending only on $d, \nu, K, b - a, p, c_0, \varepsilon_0$ and $\alpha$. Taking $W_{\min}$ large enough, the first term $\varepsilon_1$ is always larger than the second term in the above minimum, and the logarithmic terms $\log(W \log(W))$ and $\log(LW \log(W))$ can be upper bounded by a constant times $\log(W)$ (since $L \leq W$). Rearranging concludes the proof. □

## C Earlier works: two other lower bound proof strategies

Approximation lower bounds in a sense similar to ours have been obtained in other recent works. In the purpose of highlighting the differences between the different approaches, we describe the lower bound proof strategies of Yarotsky [Yar17] and of Petersen and Voigtlaender [PV18].

### C.1 Approximation in sup norm of Sobolev unit balls with ReLU networks [Yar17]

Recall that the Sobolev space $\mathcal{W}^{n,\infty}([0,1]^d)$ is defined as the set of functions on $[0,1]^d$ lying in $L^\infty$ along with all their weak derivatives up to order $n$. We equip this space with the norm

$$\|f\|_{\mathcal{W}^{n,\infty}([0,1]^d)} = \max_{\mathbf{n} \in \mathbb{N}^d : |\mathbf{n}| \leq n} \mathrm{ess\,sup}_{x \in [0,1]^d} |D^{\mathbf{n}} f(x)|,$$

and we let $F_{n,d}$ be the unit ball of this space.

We first state the sup norm lower bound and then we give a synthesized version of the proof.

**Proposition 8** ([Yar17]). *There exists positive constants $W_{\min}, c > 0$ such that for any feed-forward neural network with architecture $\mathcal{A}$, ReLU activation and $W \geq W_{\min}$ weights,*

$$\sup_{f \in F_{n,d}} \inf_{g \in H_{\mathcal{A}}} \|f - g\|_\infty \geq cW^{-\frac{2n}{d}} \,.$$

Details aside, the proof reads as follows. The author assumes that $H_{\mathcal{A}}$ approximates $F_{n,d}$ with error $\varepsilon$. Fixing $N = c_{n,d}(3\varepsilon)^{-1/n}$ for a properly chosen constant $c_{n,d} > 0$, he constructs a set of functions in $F_{n,d}$ that can shatter a grid of $N^d$ points $x_1, \ldots, x_{N^d}$ evenly distributed over $[0,1]^d$. The assumption that $H_{\mathcal{A}}$ approximates $F_{n,d}$ in sup norm with error $\varepsilon$ allows to conclude that $H_{\mathcal{A}}$ also shatters $\{x_1, \ldots, x_{N^d}\}$, and hence, $\mathrm{VCdim}(H_{\mathcal{A}}) \geq N^d = c'_{n,d} \varepsilon^{-\frac{d}{n}}$, for a properly chosen constant

---

[7]This is because $\mathrm{Pdim}(H_{\mathcal{A}}) = \mathrm{VCdim}\big(\{(x, r) \in \mathbb{R}^d \times \mathbb{R} \mapsto \mathbb{1}_{\{g(x) - r > 0\}} : g \in H_{\mathcal{A}}\}\big)$, the output neuron of $\mathcal{A}$ is linear, and we allow skip connections.

$c'_{n,d} > 0$. The author concludes using the upper bound on $\text{VCdim}(H_{\mathcal{A}})$ with respect to $W$ from [AB99] which yields $\text{VCdim}(H_{\mathcal{A}}) \leq c'W^2$ for some constant $c'$.

It is worth stressing that in this proof, it is paramount to assume that $H_{\mathcal{A}}$ approximates $F_{n,d}$ in sup norm, rather than any $L^p$ norm with $p < +\infty$. The reason is that only this choice of norm allows to bound the discrepancy between $f \in F_{n,d}$ and $g_f \in H_{\mathcal{A}}$ chosen optimally with respect to $f$ at any chosen points. Our proof strategy relying on Proposition 1 allows to circumvent this issue by relating the pseudo-dimension to the metric entropy with respect to any $L^p$ norm, $1 \leq p < +\infty$.

### C.2  Approximation in $L^p$ norm of *Horizon functions* with quantized networks [PV18]

The authors study *quantized* neural networks, that is, networks with weights constrained to be representable with a fixed number of bits. They obtain a lower bound on the minimal number of weights in a quantized neural network that can approximate a set of *Horizon functions* in $L^p$ norm, $p > 0$, with error $\varepsilon > 0$. This lower bound is easily invertible to a bound on the approximation error and is thus comparable to the results we obtain in this paper.

Textually, the authors introduce the set of horizon functions as follows: "These are $\{0, 1\}$-valued functions with a jump along a hypersurface and such that the jump surface is the graph of a smooth function" [PV18]. Denoting by $H$ the indicator function of the set $[0, +\infty) \times \mathbb{R}^{d-1}$, the set of horizon functions reads as

$$\mathcal{HF}_{\beta,d,B} = \left\{ f \circ T \in L^\infty \left( \left[ -\frac{1}{2}, \frac{1}{2} \right]^d \right) \ : \right.$$

$$\left. f(x) = H(x_1 + \gamma(x_2, \ldots, x_d), x_2, \ldots, x_d), \gamma \in \mathcal{F}_{\beta,d-1,B}, \ T \in \Pi(d, \mathbb{R}) \right\},$$

where $\mathcal{F}_{\beta,d-1,B}$ denotes the set of Hölder functions over $[-1/2, 1/2]^{d-1}$ whith smoothness parameter $\beta$ and with norm $\|.\|_{\mathcal{C}^{n,\alpha}}$ bounded by $B$ (see Section 3), and $\Pi(d, \mathbb{R})$ denotes the group of $d$-dimensional permutation matrices.

In the following, for any nonzero integer $K$ and any neural network architecture $\mathcal{A}$, we denote by $H_{\mathcal{A}}^K \subset H_{\mathcal{A}}$ the set of $K$-quantized functions in $H_{\mathcal{A}}$; namely, the functions in $H_{\mathcal{A}}$ with weights representable using at most $K$ bits. The lower bound in [PV18] (Theorem 4.2) reads as follow:

**Proposition 9** ([PV18]). *Let $d \geq 2$. Let $p, \beta, B, c_0 > 0$ and let $\sigma : \mathbb{R} \to \mathbb{R}$ be such that $\sigma(0) = 0$. There exist positive constants $\varepsilon_0, c > 0$ depending only on $d, p, \beta, B$ and $c_0$ such that, for any $\varepsilon \leq \varepsilon_0$, setting $K = \lceil c_0 \log(1/\varepsilon) \rceil$, for any feed-forward neural network architecture $\mathcal{A}$ with $W$ weights and activation $\sigma$ such that $H_{\mathcal{A}}^K$ approximates $\mathcal{HF}_{\beta,d,B}$ in $L^p$ norm with error less than $\varepsilon$, we have*

$$W \geq c\varepsilon^{-\frac{p(d-1)}{\beta}} \log^{-1}(1/\varepsilon).$$

The proof of this result is based on a lemma giving a lower bound on the minimal number of bits $\ell$ necessary for a binary encoder-decoder pair to achieve an error less than $\varepsilon > 0$ in approximating $\mathcal{HF} := \mathcal{HF}_{\beta,d,B}$ in $L^p$ norm. Formally, given an integer $\ell > 0$, a binary encoder $E^\ell : \mathcal{HF} \to \{0, 1\}^\ell$ and given a decoder $D^\ell : \{0, 1\}^\ell \to \mathcal{HF}$, one can measure an approximation error

$$\sup_{f \in \mathcal{HF}} \|f - D^\ell(E^\ell(f))\|_{L^p},$$

which quantifies the loss of information due to the encoding $E^\ell$. Clearly, for an optimal choice of encoder, one can reduce this loss of information by increasing $\ell$. In particular, for $\varepsilon > 0$, it is possible to estimate

$$\ell_\varepsilon = \min \left\{ \ell > 0 \ : \ \inf_{E^\ell, D^\ell} \sup_{f \in \mathcal{HF}} \|f - D^\ell(E^\ell(f))\|_{L^p} \leq \varepsilon \right\},$$

with the convention that $\ell_\varepsilon = \infty$ if the above set is empty. The authors show in their Lemma B.3 that for $\varepsilon$ small enough (smaller than some $\varepsilon_0 > 0$), it holds that

$$\ell_\varepsilon \geq c\varepsilon^{-\frac{p(d-1)}{\beta}} \tag{24}$$

for some constant $c > 0$ depending only on $d, p, \beta$ and $B$. In other words, one can not achieve a loss of information smaller than $\varepsilon$ by encoding functions in $\mathcal{HF}$ over less than $c\varepsilon^{-\frac{p(d-1)}{\beta}}$ bits.

The rest of the proof consists in showing that for an integer $K > 0$, given a neural network architecture $\mathcal{A}$ with $W$ weight that can approximate $\mathcal{HF}$ in $L^p$ norm with error less than $\varepsilon > 0$, one can encode exactly (without loss of information, and for a given activation function) any function in $H_{\mathcal{A}}^K$ over a string of $\ell = c_1 W(K + \lceil \log_2 W \rceil)$ bits. This generates a natural encoder-decoder system where any function $f \in \mathcal{HF}$ is encoded as the bit string of length $\ell$ associated to $g_f \in H_{\mathcal{A}}^K$ chosen to approximate $f$. It remains to observe that if we fix $K$, this automatically yields a lower bound on $\ell$ using inequality (24), and thus on $W$ by expressing $W$ through $\ell$ and $K$.

**Remark.** The authors in [PV18] study the neural network approximation in a setting slightly different from ours, since they focus on the approximation by quantized neural networks. This partly explains why their proof strategy differs from ours. However, it is worth pointing out that the proof of their lower bound on the minimal number of bits required to accurately encode a function in $\mathcal{HF}$ relies on a lower bound of the packing number of $\mathcal{HF}$, just like the lower bound of the packing number of the set to approximate is key in our proof strategy. An interesting question for the future would be to see whether our general lower bound (Theorem 1) yields lower bounds of the same order as those in [PV18] for quantized neural networks.

# D   Hölder balls

## D.1   Proof of Lemma 2

Though not necessarily stated this way, many arguments below are classical (see, e.g., Theorem 3.2 by [GKKW02] with a similar construction for lower bounds in nonparametric regression).

Let $N \in \mathbb{N}^*$. For $\mathbf{m} = (m_1, \dots, m_d) \in \{0, \dots, N-1\}^d$, we let $x_{\mathbf{m}} := \frac{1}{N}(m_1 + 1/2, \dots, m_d + 1/2)$ and we denote by $C_{\mathbf{m}}$ the cube of side-length $\frac{1}{N}$ centered at $x_{\mathbf{m}}$, with sides parallel to the axes. We see that the $N^d$ cubes $C_{\mathbf{m}}$ decompose the cube $[0,1]^d$ in smaller parts which, up to negligible sets which will not be problematic, form a partition of $[0,1]^d$. We will use this decomposition to construct a packing of $F_{s,d}$. Denoting $\|\cdot\|$ the sup norm in $\mathbb{R}^d$, we define the $C^\infty$ test function $\phi : \mathbb{R}^d \to \mathbb{R}$ by:

$$\phi(x) = \exp\left(-\frac{\|x\|^2}{1 - \|x\|^2}\right),$$

for any $x \in \mathbb{R}^d$ such that $\|x\| < 1$, and $\phi(x) = 0$ otherwise. Recalling that $n \in \mathbb{N}$ and $\alpha \in (0,1]$ are such that $s = n + \alpha$, and since all the high-order partial derivatives of $\phi$ are uniformly bounded on $[0,1]^d$, $\|\phi\|_{\mathcal{C}^{n,\alpha}}$ is thus finite and is nonzero.

Let $c_s = \frac{1}{2}(2N)^{-s}\|\phi\|_{\mathcal{C}^{n,\alpha}}^{-1}$ and consider, for any tensor of signs $\sigma = (\sigma_{\mathbf{m}})_{\mathbf{m} \in \{0,\cdots,N-1\}^d} \in \{-1,1\}^{N^d}$, the function $f_\sigma$ defined as follows:

$$f_\sigma(x) = c_s \sum_{\mathbf{m} \in \{0,\dots,N-1\}^d} \sigma_{\mathbf{m}}\phi\left(2N(x - x_{\mathbf{m}})\right),$$

for all $x \in [0,1]^d$. There are $2^{N^d}$ different functions $f_\sigma$.

Let us prove that, for all $\sigma \in \{-1,1\}^{N^d}$, $f_\sigma \in F_{s,d}$. To do so, we study the constituents of $\|f_\sigma\|_{\mathcal{C}^{n,\alpha}}$ separately and show that they are all bounded by 1. For $\mathbf{m} \in \{0,\cdots,N-1\}^d$, we define the function $g_{\mathbf{m}}(x) = c_s\sigma_{\mathbf{m}}\phi\left(2N(x - x_{\mathbf{m}})\right)$. Note that because $\phi$ vanishes outside $(-1,1)^d$, we have that $g_{\mathbf{m}}$ vanishes everywhere outside the interior of $C_{\mathbf{m}}$, and the same holds for $D^{\mathbf{n}}g_{\mathbf{m}}$ for all $\mathbf{n} \in \mathbb{N}^d$ such that $|\mathbf{n}| \leq n$. For any such $\mathbf{n}$, we have

$$\|D^{\mathbf{n}}g_{\mathbf{m}}\|_\infty = c_s(2N)^{|\mathbf{n}|}\|D^{\mathbf{n}}\phi\|_\infty \leq c_s(2N)^s\|\phi\|_{\mathcal{C}^{n,\alpha}} \leq \frac{1}{2}.$$

Therefore,

$$\max_{\mathbf{n}:|\mathbf{n}|\leq n} \|D^{\mathbf{n}}f_\sigma\|_\infty \leq 1.$$

Now for any $\mathbf{n} \in \mathbb{N}^d$ such that $|\mathbf{n}| = n$, any $x, y \in [0,1]^d$, we have
$$\frac{|D^{\mathbf{n}}f_\sigma(x) - D^{\mathbf{n}}f_\sigma(y)|}{\|x - y\|_2^\alpha} = \frac{|D^{\mathbf{n}}g_{\mathbf{m}}(x) - D^{\mathbf{n}}g_{\mathbf{m}'}(y)|}{\|x - y\|_2^\alpha},$$
where $x \in C_{\mathbf{m}}$ and $y \in C_{\mathbf{m}'}$ for some multi-indexes $\mathbf{m}$ and $\mathbf{m}'$. We have to distinguish between the cases $\mathbf{m} = \mathbf{m}'$ and $\mathbf{m} \neq \mathbf{m}'$. In the former case, we have
$$\begin{aligned}\frac{|D^{\mathbf{n}}f_\sigma(x) - D^{\mathbf{n}}f_\sigma(y)|}{\|x - y\|_2^\alpha} &= c_s(2N)^{n+\alpha}\frac{|D^{\mathbf{n}}\phi(2N(x - x_{\mathbf{m}})) - D^{\mathbf{n}}\phi(2N(y - x_{\mathbf{m}}))|}{\|2N(x - x_{\mathbf{m}}) - 2N(y - x_{\mathbf{m}})\|_2^\alpha}\\ &= c_s(2N)^s\frac{|D^{\mathbf{n}}\phi(x') - D^{\mathbf{n}}\phi(y')|}{\|x' - y'\|_2^\alpha}\\ &\leq c_s(2N)^s\|\phi\|_{\mathcal{C}^{n,\alpha}} = \frac{1}{2},\end{aligned}$$
where at the second line, we used the changes of variables $x' = 2N(x - x_{\mathbf{m}})$ and $y' = 2N(y - x_{\mathbf{m}})$. In the case $\mathbf{m} = \mathbf{m}'$ ($x$ and $y$ belong to the same cube), we thus have
$$\frac{|D^{\mathbf{n}}f_\sigma(x) - D^{\mathbf{n}}f_\sigma(y)|}{\|x - y\|_2^\alpha} \leq 1.$$

In the case $\mathbf{m} \neq \mathbf{m}'$, observe that we have
$$|D^{\mathbf{n}}g_{\mathbf{m}}(x) - D^{\mathbf{n}}g_{\mathbf{m}'}(y)| \leq 2\max\{|D^{\mathbf{n}}g_{\mathbf{m}}(x)|, |D^{\mathbf{n}}g_{\mathbf{m}'}(y)|\}. \tag{25}$$

Besides, recall that $D^{\mathbf{n}}g_{\mathbf{m}}$ and $D^{\mathbf{n}}g_{\mathbf{m}'}$ both vanish outside of the interiors of $C_{\mathbf{m}}$ and $C_{\mathbf{m}'}$ respectively. We can thus rewrite (25) as
$$\begin{aligned}|D^{\mathbf{n}}g_{\mathbf{m}}(x) - D^{\mathbf{n}}g_{\mathbf{m}'}(y)| &\leq 2\max\{|D^{\mathbf{n}}g_{\mathbf{m}}(x) - D^{\mathbf{n}}g_{\mathbf{m}}(y)|, |D^{\mathbf{n}}g_{\mathbf{m}'}(x) - D^{\mathbf{n}}g_{\mathbf{m}'}(y)|\}\\ &\leq 2c_s(2N)^n\max\{|D^{\mathbf{n}}\phi(2N(x - x_{\mathbf{m}})) - D^{\mathbf{n}}\phi(2N(y - x_{\mathbf{m}}))|,\\ &\qquad\qquad\qquad |D^{\mathbf{n}}\phi(2N(y - x_{\mathbf{m}'})) - D^{\mathbf{n}}\phi(2N(y - y_{\mathbf{m}'}))|\}.\end{aligned}$$

This entails
$$\begin{aligned}\frac{|D^{\mathbf{n}}f_\sigma(x) - D^{\mathbf{n}}f_\sigma(y)|}{\|x - y\|_2^\alpha} &\leq c_s2(2N)^s\max\left\{\frac{|D^{\mathbf{n}}\phi(x') - D^{\mathbf{n}}\phi(y')|}{\|x' - y'\|_2^\alpha}, \frac{|D^{\mathbf{n}}\phi(x'') - D^{\mathbf{n}}\phi(y'')|}{\|x'' - y''\|_2^\alpha}\right\}\\ &\leq c_s2(2N)^s\|\phi\|_{\mathcal{C}^{n,\alpha}} = 1,\end{aligned}$$
where $x' = 2N(x - x_{\mathbf{m}})$ and $y' = 2N(y - x_{\mathbf{m}})$, and $x'' = 2N(x - x_{\mathbf{m}'})$ and $y'' = 2N(y - x_{\mathbf{m}'})$.

Summarizing, we showed that for all $\sigma \in \{-1, 1\}^{N^d}$
$$\max_{\mathbf{n}:|\mathbf{n}|\leq n}\|D^{\mathbf{n}}f_\sigma\|_\infty \leq 1 \qquad \text{and} \qquad \max_{\mathbf{n}:|\mathbf{n}|=n}\sup_{x\neq y}\frac{|D^{\mathbf{n}}f_\sigma(x) - D^{\mathbf{n}}f_\sigma(y)|}{\|x - y\|_2^\alpha} \leq 1.$$

We conclude that for all $\sigma \in \{-1, 1\}^{N^d}$
$$\|f_\sigma\|_{\mathcal{C}^{n,\alpha}} \leq 1,$$
and therefore $\{f_\sigma : \sigma \in \{-1, 1\}^{N^d}\} \subset F_{s,d}$.

Let us now evaluate the distance between distinct elements of $\{f_\sigma : \sigma \in \{-1, 1\}^{N^d}\}$. Let $\sigma^1$, $\sigma^2 \in \{-1, 1\}^{N^d}$, with $\sigma^1 \neq \sigma^2$, and let $\mathbf{m} \in \{0, \ldots, N - 1\}^d$ be such that $\sigma_{\mathbf{m}}^1 = -\sigma_{\mathbf{m}}^2$. Let us estimate $\Delta_p$ the $L^p(\lambda)$ discrepancy between $f_{\sigma^1}$ and $f_{\sigma^2}$ on the cube $C_{\mathbf{m}}$, that is
$$\begin{aligned}\Delta_p^p &= \int_{C_{\mathbf{m}}}|f_{\sigma^1}(x) - f_{\sigma^2}(x)|^p\mathrm{d}x\\ &= 2^pc_s^p\int_{C_{\mathbf{m}}}|\phi(2N(x - x_{\mathbf{m}}))|^p\mathrm{d}x\\ &= 2^pc_s^p(2N)^{-d}\|\phi\|_{L^p(\lambda)}^p.\end{aligned}$$

It remains to find a subset among the functions $f_\sigma$ such that any two functions of this set differ on a significant number of cubes $C_{\mathbf{m}}$. According to the Varshamov-Gilbert Lemma [Yu97], there exists

$\Gamma \subset \{-1, 1\}^{N^d}$ with cardinal at least $\exp(N^d/8)$ such that for any $\sigma^1, \sigma^2 \in \Gamma$, such that $\sigma^1 \neq \sigma^2$, $\sigma^1$ and $\sigma^2$ differ on at least one fourth of their coordinates; i.e., $\sum_{k=1}^{N^d} \mathbb{1}_{\sigma_k^1 \neq \sigma_k^2} \geq \frac{N^d}{4}$. We thus fix such a set $\Gamma \subset \{-1, 1\}^{N^d}$. For any $\sigma^1, \sigma^2 \in \Gamma$, with $\sigma^1 \neq \sigma^2$,

$$
\begin{aligned}
\|f_{\sigma^1} - f_{\sigma^2}\|_{L^p(\lambda)}^p &= \sum_{\mathbf{m}:\sigma_\mathbf{m}^1 \neq \sigma_\mathbf{m}^2} \int_{C_\mathbf{m}} |f_{\sigma^1}(x) - f_{\sigma^2}(x)|^p \mathrm{d}x \\
&\geq \frac{N^d}{4}\Delta_p^p = \frac{2^{p-d}c_s^p}{4}\|\phi\|_{L^p(\lambda)}^p.
\end{aligned}
$$

Finally, recalling the definition of $c_s$, we have for any $\sigma^1, \sigma^2 \in \Gamma$, with $\sigma^1 \neq \sigma^2$,

$$
\|f_{\sigma^1} - f_{\sigma^2}\|_{L^p(\lambda)} \geq 2^{1-\frac{d+2}{p}}\frac{1}{2}(2N)^{-s}\|\phi\|_{\mathcal{C}^{n,\alpha}}^{-1}\|\phi\|_{L^p(\lambda)} = cN^{-s},
$$

where $c = 2^{-s-\frac{d+2}{p}}\frac{\|\phi\|_{L^p(\lambda)}}{\|\phi\|_{\mathcal{C}^{n,\alpha}}}$.

It follows that $\{f_\sigma : \sigma \in \Gamma\}$ is a $cN^{-s}$-packing of $F_{s,d}$. Given the lower bound on the size of $\Gamma$, this implies

$$
M\left(cN^{-s}, F_{s,d}, \|\cdot\|_{L^p(\lambda)}\right) \geq \exp(N^d/8),
$$

for all $N \in \mathbb{N}^*$.

Set $\varepsilon_0 = c$ and $c_0 = 2^{-d}c^{\frac{d}{s}}/8$. Consider $\varepsilon > 0$, with $\varepsilon \leq \varepsilon_0$. To conclude the proof, we need to show that (12) holds for $\varepsilon$. To do so, we consider $N$: the smallest integer such that $cN^{-s} \geq \varepsilon \geq c(2N)^{-s}$. This $N \in \mathbb{N}^*$ exists because $0 < \varepsilon \leq \varepsilon_0 = c$ and $s > 0$. On one side, we have

$$
M\left(\varepsilon, F_{s,d}, \|\cdot\|_{L^p(\lambda)}\right) \geq M\left(cN^{-s}, F_{s,d}, \|\cdot\|_{L^p(\lambda)}\right),
$$

and on the other side, since $2N \geq c^{\frac{1}{s}}\varepsilon^{-\frac{1}{s}}$,

$$
\exp(N^d/8) \geq \exp(2^{-d}c^{\frac{d}{s}}\varepsilon^{-\frac{d}{s}}/8) = \exp(c_0\varepsilon^{-\frac{d}{s}}).
$$

Combining the last three inequalities, we finally obtain

$$
\log M\left(\varepsilon, F_{s,d}, \|\cdot\|_{L^p(\lambda)}\right) \geq c_0\varepsilon^{-d/s},
$$

for all $0 < \varepsilon \leq \varepsilon_0$.

# E   Monotonic functions

This section contains the proofs of the results stated in Section 4. More precisely, in Section E.1 we provide the proof of Proposition 6 and in Section E.2 we provide the proof of Proposition 4.

## E.1   Proof of Proposition 6

The section contains two sub-sections. In the first sub-section, we provide a proposition on the representation of piecewise-constant functions with Heaviside neural-networks. Section E.1.2 contains the main part of the proof of Proposition 6.

### E.1.1   Representing piecewise-constant functions with Heaviside neural networks

We first describe a neural network architecture which, with the Heaviside activation function, is able to represent functions that are piecewise-constant on cubes.

**Proposition 10.** *Let $d \in \mathbb{N}^*$, $M \in \mathbb{N}^*$. There exists an architecture $\mathcal{A}$ with two-hidden layers, $2(d+1)^2M$ weights and the Heaviside activation function, such that for any $(\alpha_i)_{1 \leq i \leq M} \in \mathbb{R}^M$, any collection $(\mathcal{C}_i)_{1 \leq i \leq M}$ of mutually disjoint hypercubes of $\mathbb{R}^d$ the function $\tilde{f}: \mathbb{R}^d \to [0, 1]$ defined by*

$$
\forall x \in \mathbb{R}^d, \quad \tilde{f}(x) = \sum_{i=1}^M \alpha_i \mathbb{1}_{\mathcal{C}_i}(x)
$$

*satisfies $\tilde{f} \in H_\mathcal{A}$.*

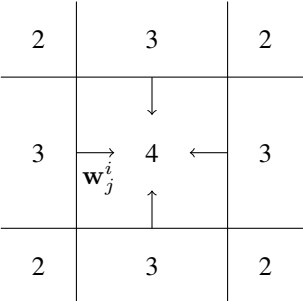

Figure 1: Values of the sum of the perceptrons $p_j^i(x)$ around a hypercube $\mathcal{C}_i$ in dimension 2.

*Proof.* Define $\sigma\colon \mathbb{R} \to \mathbb{R}$ by $\sigma(x) = \mathbb{1}_{x \geq 0}$ for all $x \in \mathbb{R}$.

Let $i \in \{1, \ldots, M\}$. The cube $\mathcal{C}_i$ has $2d$ faces. These faces are supported by hyperplanes whose equations are of the form $\langle \mathbf{w}, x \rangle + b = 0$, with $\mathbf{w} \in \mathbb{R}^d$ and $b \in \mathbb{R}$. We allow the faces to belong to the cube or not. To distinguish them, we denote $J_i \in \{1, \ldots, 2d\}$ the number of faces that belong to $\mathcal{C}_i$. We index the $J_i$ faces that belong to the cube from 1 to $J_i$, and the other faces from $J_i + 1$ to $2d$. Thus, for all $i \in \{1, \ldots, M\}$ and all $j \in \{1, \ldots, 2d\}$, there exist $\mathbf{w}_j^i \in \mathbb{R}^d, b_j^i \in \mathbb{R}$ such that

$$\mathcal{C}_i = \bigcap_{j=1}^{J_i} \{x \in \mathbb{R}^d\colon \langle \mathbf{w}_j^i, x \rangle + b_j^i \geq 0\} \cap \bigcap_{j=J_i+1}^{2d} \{x \in \mathbb{R}^d\colon \langle \mathbf{w}_j^i, x \rangle + b_j^i > 0\}.$$

We rewrite:

$$\mathcal{C}_i = \left\{ x \in \mathbb{R}^d\colon \sum_{j=1}^{J_i} \mathbb{1}_{\{\langle \mathbf{w}_j^i, x \rangle + b_j^i \geq 0\}} + \sum_{j=J_i+1}^{2d} \mathbb{1}_{\{\langle \mathbf{w}_j^i, x \rangle + b_j^i > 0\}} \geq 2d \right\}. \tag{26}$$

Denoting for all $i \in \{1, \ldots, M\}$ and all $j \in \{1, \ldots, 2d\}$ and for all $x \in \mathbb{R}^d$,

$$p_j^i(x) = \begin{cases} \sigma\left(\langle \mathbf{w}_j^i, x \rangle + b_j^i\right) & \text{if } j \leq J_i \\ 1 - \sigma\left(-\langle \mathbf{w}_j^i, x \rangle - b_j^i\right) & \text{otherwise,} \end{cases}$$

we have, see Figure 1 and (26), for all $x \in \mathbb{R}^d$

$$\mathbb{1}_{\mathcal{C}_i}(x) = \begin{cases} 1 & \text{if } \sum_{j=1}^{2d} p_j^i(x) \geq 2d, \\ 0 & \text{otherwise,} \end{cases}$$

$$= \sigma\left(\sum_{j=1}^{2d} p_j^i(x) - 2d\right).$$

Since the hypercubes are mutually disjoints, for all $x \in [0, 1]^d$, we have

$$\tilde{f}(x) = \sum_{i=1}^{M} \alpha_i \sigma\left(\sum_{j=1}^{J_i} \sigma\left(\langle \mathbf{w}_j^i, x \rangle + b_j^i\right) + \sum_{j=J_i+1}^{2d} \left(1 - \sigma\left(-\langle \mathbf{w}_j^i, x \rangle - b_j^i\right)\right) - 2d\right)$$

$$= \sum_{i=1}^{M} \alpha_i \sigma\left(\sum_{j=1}^{2d} \varepsilon_j^i \sigma\left(\langle \tilde{\mathbf{w}}_j^i, x \rangle + \tilde{b}_j^i\right) - J_i\right), \tag{27}$$

where

$$\varepsilon_j^i = \begin{cases} +1 & \text{if } j \leq J_i \\ -1 & \text{otherwise,} \end{cases} \qquad \tilde{\mathbf{w}}_j^i = \begin{cases} \mathbf{w}_j^i & \text{if } j \leq J_i \\ -\mathbf{w}_j^i & \text{otherwise,} \end{cases} \qquad \tilde{b}_j^i = \begin{cases} b_j^i & \text{if } j \leq J_i \\ -b_j^i & \text{otherwise.} \end{cases}$$

Equation (27) is the action of the Heaviside neural network with two hidden layers whose architecture is on Figure 2.

It remains to count the weights and biases of $\tilde{f}$ :

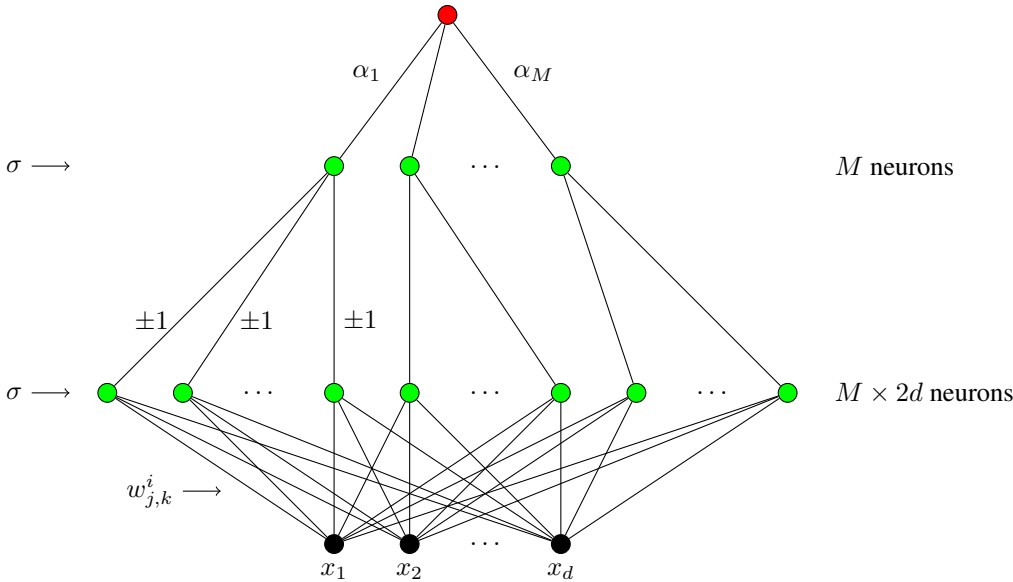

Figure 2: The function $\tilde{f}$ represented as a neural network.

- the architecture has $M$ edges going to the output layer, due to the $\alpha_i$;

- it has $M$ biases associated to the neurons of the second hidden layer (they correspond to the terms $-J_i$);

- between the second and the first hidden layer, the architecture has $M \times 2d$ edges (corresponding to the $\varepsilon_{i,j}$);

- it has $M \times 2d$ biases associated to the neurons of the first hidden layer (the $\tilde{b}_j^i$);

- it has $M \times 2d \times d$ edges between the first hidden layer and the entry (the $\tilde{\mathbf{w}}_j^i$).

Thus there are $2M + 2M \times 2d + M \times 2d \times d = 2(d^2 + 2d + 1)M = 2(d+1)^2 M$ weights and biases in total. $\qquad\qquad\qquad\qquad\qquad\qquad\qquad\qquad\qquad\qquad\qquad\qquad\qquad\qquad\qquad\square$

### E.1.2  Main developments of the proof of Proposition 6

Let $N \in \mathbb{N}^*$ and $f \in \mathcal{M}^d$. In this section, we partition $[0,1)^d$ into cubes whose sizes depend on the maximal variation of $f$. Then we use this partition to construct a piecewise constant approximation $\tilde{f}$ of $f$; we will bound from above the $L^p(\lambda)$ approximation error $\|f - \tilde{f}\|_{L^p(\lambda)}$ by a function of $N$. This part is a direct reinterpretation of the proof of Proposition 3.1 in [GW07]. We then apply Proposition 10 to $\tilde{f}$ and obtain the announced result.

We first define some notation that will be used in the rest of the section, then we explain the algorithm used to divide $[0,1)^d$ into cubes. We fix the constant $K > 1$ the following way:

$$K := \begin{cases} 2^d & \text{if } p = 1, \\ 2^\beta & \text{otherwise, where } \beta = \frac{1}{2}(d - 1 + \frac{1}{p-1}). \end{cases}$$

We also define an integer $l$ that corresponds to the number of cube decompositions:

$$l := \left\lceil \frac{N \log 2}{\log K} \right\rceil = \begin{cases} \left\lceil \frac{N}{d} \right\rceil & \text{if } p = 1, \\ \left\lceil \frac{N}{\beta} \right\rceil & \text{otherwise .} \end{cases} \tag{28}$$

It is worth noting that this implies $K^{-l} \le 2^{-N} < K^{-l+1}$.

Now we partition $[0, 1)^d$ into dyadic cubes of the form $[a_1, b_1) \times \cdots \times [a_d, b_d)$. If $C$ is such a cube, we use the following convenient notation:

$$\underline{C} := (a_1, \ldots, a_d) \in \mathbb{R}^d, \qquad\qquad \overline{C} := (b_1, \ldots, b_d) \in \mathbb{R}^d,$$

to refer to the smallest and largest vertices of $C$. The cube decompositions process reads as follow:

- First we partition $[0, 1)^d$ into $2^{Nd}$ cubes of side-length $2^{-N}$. We denote by $S_0$ the set of these cubes $C$ such that $f(\overline{C}) - f(\underline{C}) \leq K2^{-N}$ and by $R_0$ the set of the remaining cubes.

- For $1 \leq i < l$, we partition each cube in the set $R_{i-1}$ (the remaining cubes at the step $i-1$) into $2^d$ cubes of equal size, and we denote by $S_i$ the set of obtained cubes $C$ of side-length $2^{-(i+N)}$ such that

$$f(\overline{C}) - f(\underline{C}) \leq K^{i+1}2^{-N}. \tag{29}$$

  Again, the set of remaining cubes is denoted by $R_i$.

- Lastly, we partition each cube in the set $R_{l-1}$ into $2^d$ cubes of equal size, and we denote by $S_l$ the set of obtained cubes of side-length $2^{-(l+N)}$.

Once the algorithm is done, each point in $[0, 1)^d$ clearly belongs to one single cube of $\cup_{i=0}^l S_i$. For $i \in \{0, \ldots, l\}$, we let $\tilde{S}_i = \cup_{C \in S_i} C$.

We now define the piecewise constant approximation of $f$ by

$$\forall x \in [0, 1]^d, \quad \tilde{f}(x) = \sum_{C \in \cup_{0 \leq i \leq l} S_i} f(\underline{C}) \mathbb{1}_{x \in C},$$

where $\mathbb{1}_{x \in C}$ denotes the indicator function of the cube $C$. We do not make the dependence explicit, but $\tilde{f}$ depends on the parameters $N$, $d$ and $p$. The number of cubes over which $\tilde{f}$ is constant is $\sum_{i=0}^l |S_i|$. This quantity is key when constructing the neural network according to Proposition 10; in the next lemma, we bound from above $|S_i|$ for all $i = 0, \ldots, l$. Then, we will estimate the error $\|f - \tilde{f}\|_{L^p(\lambda)}$.

**Lemma 7.** *With the above notation:*

$$\forall i \in \{0, \ldots, l\}, \quad |S_i| \leq dK^{-i}2^{i(d-1)+Nd+1}$$

*Moreover,*

$$\lambda(\tilde{S}_i) \leq \begin{cases} 1 & \text{if } i = 0, \\ 2d(2K)^{-i} & , \text{otherwise}. \end{cases} \tag{30}$$

*Proof.* By construction, we have

$$\forall i \in \{1, \ldots, l\}, \quad |S_i| + |R_i| = 2^d|R_{i-1}|,$$

since the set $S_i \cup R_i$ contains all the cubes of side-length $2^{-(i+N)}$, that have been constructed from the cubes of $R_{i-1}$. In particular,

$$\forall i \in \{1, \ldots, l\}, \quad |S_i| \leq 2^d|R_{i-1}|. \tag{31}$$

It remains to bound $|R_{i-1}|$ from above for $i \geq 1$. Define $V := \{\underline{C} : C \in R_{i-1}\}$ the set of the smallest vertices of the cubes in $R_{i-1}$. We consider the classes of these vertices under the "laying on the same extended diagonal" equivalence relation. Since the cubes have side-length $2^{-(i-1+N)}$, there are less than $d2^{(i-1+N)(d-1)}$ equivalence classes. According to the pigeonhole principle, the largest class has at least $\left\lceil \frac{|V|}{d2^{(i-1+N)(d-1)}} \right\rceil$ elements; let us refer to this class as $\mathcal{D}$. Let $(C_j)_{1 \leq j \leq J}$ be the set of cubes in $R_{i-1}$ having a point in $\mathcal{D}$ as lowest vertex. Since $f$ is non-decreasing and according to (29), we have:

$$1 \geq f(1, \ldots, 1) - f(0, \ldots, 0) \geq \sum_{j=1}^J f(\overline{C_j}) - f(\underline{C_j}) \geq JK^i2^{-N}$$

$$\geq \frac{|V|}{d2^{(i-1+N)(d-1)}}K^i2^{-N} = \frac{|R_{i-1}|}{d2^{(i-1+N)(d-1)}}K^i2^{-N}.$$

Thus
$$|R_{i-1}| \le d2^{i(d-1)+Nd+1-d}K^{-i}.$$
The first statement of Lemma 7 follows from (31).

For $i = 0$, $\lambda(\tilde{S}_0) \le 1$. For $1 \le i \le l$, using the first statement of this lemma, we bound from above the measure of $\tilde{S}_i$:

$$\lambda(\tilde{S}_i) = \left(2^{-(i+N)}\right)^d |S_i| \quad \le \quad dK^{-i}2^{i(d-1)+Nd+1}2^{-d(i+N)},$$
$$= \quad 2d(2K)^{-i}.$$

$\square$

To show that $\tilde{f}$ is close to $f$ in $L^p(\lambda)$ norm, let us use the fact that $(\tilde{S}_i)_{0 \le i \le l}$ is a partition of $[0, 1)^d$ and decompose the error in three parts:

$$\|f - \tilde{f}\|^p_{L^p(\lambda)} = \int_{\tilde{S}_0} |f(x) - \tilde{f}(x)|^p \mathrm{d}x + \sum_{i=1}^{l-1} \int_{\tilde{S}_i} |f(x) - \tilde{f}(x)|^p \mathrm{d}x + \int_{\tilde{S}_l} |f(x) - \tilde{f}(x)|^p \mathrm{d}x.$$

In the next lemma, we control each term of the above sum to bound from above $\|f - \tilde{f}\|_{L^p(\lambda)}$ by a function of $N$ that is independent of $f$ and tends to 0 when $N$ tends to $+\infty$.

**Lemma 8.** *For any $1 \le p < +\infty$, there exists a constant $c_{d,p} > 0$ depending only on $d$ and $p$ such that for all $N \in \mathbb{N}^*$*

$$\|f - \tilde{f}\|_{L^p(\lambda)} \le c_{d,p} \begin{cases} 2^{-N} & \text{if } p(d-1) < d, \\ 2^{-N\frac{(1+1/\beta)}{p}} & \text{if } p(d-1) > d, \\ N^{\frac{1}{p}}2^{-N} & \text{if } p(d-1) = d, \end{cases} \tag{32}$$

*where $\tilde{f}$ is the function constructed for the parameters $N$, $d$ and $p$.*

*Proof.* For $0 \le i < l$, on any cube $C \in S_i$, we have

$$\forall x \in C, \quad |f(x) - \tilde{f}(x)| = |f(x) - f(\underline{C})| \le f(\overline{C}) - f(\underline{C}) \le K^{i+1}2^{-N}, \tag{33}$$

since $f$ is non-decreasing, and by definition of $\tilde{f}$ and $S_i$.

- Using the fact that $\lambda(\tilde{S}_0) \le 1$ and by (33):

$$\int_{\tilde{S}_0} |f(x) - \tilde{f}(x)|^p \mathrm{d}x \le (2^{-N}K)^p. \tag{34}$$

- Using (30) and (33), we get for all $i \in \{1, \dots, l-1\}$

$$\int_{\tilde{S}_i} |f(x) - \tilde{f}(x)|^p \mathrm{d}x \le (K^{i+1}2^{-N})^p 2d(2K)^{-i}. \tag{35}$$

- On any $C \in S_l$, we have, for all $x \in C$, $|f(x) - \tilde{f}(x)| \le |f(x) - f(\underline{C})| \le 1$, and we get, using (30):

$$\int_{\tilde{S}_l} |f(x) - \tilde{f}(x)|^p \mathrm{d}x \le 2d(2K)^{-l}. \tag{36}$$

Combining (34), (35) and (36) we get:

$$\|f - \tilde{f}\|^p_{L^p(\lambda)} \le (2^{-N}K)^p + \sum_{i=1}^{l-1} (K^{i+1}2^{-N})^p 2d(2K)^{-i} + 2d(2K)^{-l}$$

$$\le (2^{-N}K)^p + 2^{1-Np}K^p d \sum_{i=1}^{l-1} \left(\frac{K^{p-1}}{2}\right)^i + 2d(2K)^{-l}. \tag{37}$$

It remains to bound the right-hand side of (37), depending on the value of $p$ and $d$. Note that the behavior of this term depends on whether $\frac{K^{p-1}}{2}$ is larger or smaller than 1.

- Suppose that $p(d-1) < d$. In this case, we can have $p = 1$ or $p > 1$. If $p = 1$, we have $\frac{K^{p-1}}{2} = \frac{1}{2} < 1$ and $\frac{1}{2K} < K^{-p}$. If $p > 1$, we have:

$$p(d-1) < d \iff dp - p - d + 1 < 1 \iff d - 1 < \frac{1}{p-1}.$$

Thus, $\beta$ being the arithmetic mean of $d-1$ and $\frac{1}{p-1}$, we have $d - 1 < \beta < \frac{1}{p-1}$. Then $K = 2^\beta < 2^{1/(p-1)}$ and hence $\frac{K^{p-1}}{2} < 1$ and $\frac{1}{2K} < K^{-p}$. Therefore, both for $p = 1$ and $p > 1$,

$$\sum_{i=1}^{l-1} \left( \frac{K^{p-1}}{2} \right)^i \leq \frac{K^{p-1}}{2 - K^{p-1}} \quad \text{and} \quad (2K)^{-l} \leq K^{-pl}.$$

Since $K^{-l} \leq 2^{-N}$, this leads to

$$\|f - \tilde{f}\|_{L^p(\lambda)}^p \leq (2^{-N}K)^p + 2^{1-Np}K^p d \frac{K^{p-1}}{2 - K^{p-1}} + 2dK^{-pl}$$

$$\leq \left( K^p + 2K^p d \frac{K^{p-1}}{2 - K^{p-1}} + 2d \right) 2^{-Np}.$$

We thus have, setting $c_1 := \left( K^p + 2K^p d \frac{K^{p-1}}{2 - K^{p-1}} + 2d \right)^{\frac{1}{p}}$,

$$\|f - \tilde{f}\|_{L^p(\lambda)} \leq c_1 2^{-N}.$$

Notice $c_1$ only depends on $d$ and $p$.

- Suppose that $p(d-1) > d$. We have $p > 1$ and $d - 1 > \beta > \frac{1}{p-1}$. Then $K = 2^\beta > 2^{1/(p-1)}$ and hence $\frac{K^{p-1}}{2} > 1$, which entails using (37)

$$\|f - \tilde{f}\|_{L^p(\lambda)}^p \leq (2^{-N}K)^p + 2^{1-Np}K^p d \frac{(K^{p-1}/2)^l}{K^{p-1}/2 - 1} + 2d(2K)^{-l}$$

$$\leq 2^{-Np}K^p + 2^{-Np}K^{pl} \frac{2K^p d}{K^{p-1}/2 - 1}(2K)^{-l} + 2d(2K)^{-l}.$$

Since $p > 1 + \frac{1}{\beta}$, we have $2^{-Np} \leq 2^{-N(1+\frac{1}{\beta})}$. Also, since $K = 2^\beta$, $(2K)^{-l} = 2^{-l(\beta+1)}$, and since $l \geq \frac{N \log(2)}{\log(K)} = \frac{N}{\beta}$, we have $(2K)^{-l} \leq 2^{-\frac{N}{\beta}(\beta+1)} = 2^{-N(1+\frac{1}{\beta})}$. Finally, since $2^{-N}K^l < K$,

$$\|f - \tilde{f}\|_{L^p(\lambda)}^p \leq \left( K^p + K^p \frac{2K^p d}{K^{p-1}/2 - 1} + 2d \right) 2^{-N(1+1/\beta)}$$

We thus have, setting $c_2 := \left( K^p + \frac{2K^{2p}d}{K^{p-1}/2-1} + 2d \right)^{\frac{1}{p}}$,

$$\|f - \tilde{f}\|_{L^p(\lambda)} \leq c_2 2^{-\frac{N(1+1/\beta)}{p}}.$$

Notice $c_2$ only depends on $d$ and $p$.

- Suppose that $p(d-1) = d$. It implies $p > 1$ and $p - 1 = \frac{1}{d-1}$, then $\beta = d - 1$. We thus have $K^{p-1} = 2^{(d-1)(p-1)} = 2$. Therefore, (37) becomes

$$\|f - \tilde{f}\|_{L^p(\lambda)}^p \leq 2^{-Np}K^p + 2^{-Np}2K^p d(l-1) + 2d(K^p)^{-l}.$$

On the one hand, we have $K^{-l} \leq 2^{-N}$. On the other, we have $2^{-N} < K^{-l+1}$, so $l - 1 < N\frac{\log 2}{\log K} = \frac{N}{d-1}$. Putting it all together, we get

$$\|f - \tilde{f}\|_{L^p(\lambda)}^p \leq 2^{-Np}K^p + 2^{-Np}2K^p d(l-1) + 2d2^{-Np}$$

$$\leq \left( K^p + 2K^p \frac{d}{d-1} + 2d \right) N2^{-Np}.$$

We thus have, setting $c_3 := \left(K^p + 2K^p \frac{d}{d-1} + 2d\right)^{\frac{1}{p}}$,

$$\|f - \tilde{f}\|_{L^p(\lambda)} \leq c_3 N^{\frac{1}{p}} 2^{-N}.$$

Notice $c_3$ only depends on $d$ and $p$.

Letting $c_{d,p} = \max\{c_1, c_2, c_3\}$ yields the result. $\qquad\square$

According to Proposition 10, the function $\tilde{f}$ constructed for a given $N \in \mathbb{N}^*$ can be implemented by a Heaviside neural network with two hidden layers and $W = 2(d+1)^2 \sum_{i=0}^{l} |S_i|$ weights. Using Lemma 7, we obtain

$$W = 2(d+1)^2 \sum_{i=0}^{l} |S_i| \leq 2(d+1)^2 \sum_{i=0}^{l} dK^{-i} 2^{i(d-1)+Nd+1}$$

$$= 2^{Nd+2} d(d+1)^2 \sum_{i=0}^{l} \left(\frac{2^{d-1}}{K}\right)^i.$$

We let, for all $N \in \mathbb{N}^*$,

$$W_N := 2^{Nd+2} d(d+1)^2 \sum_{i=0}^{l} \left(\frac{2^{d-1}}{K}\right)^i. \tag{38}$$

Although we do not make the dependence explicit, $W_N$ also depends on $d$ and $p$. Observe that for all $d \geq 1$: $(W_N)_{N \in \mathbb{N}^*}$ is non-decreasing and $\lim_{N \to +\infty} W_N = +\infty$.

**Lemma 9.** *With the above notation: For any $+\infty > p \geq 1$, there exist constants $W'_{\min}, c'_{d,p} > 0$ depending only on $d$ and $p \geq 1$ such that for all $N$ satisfying $W_N \geq W'_{\min}$*

$$\|f - \tilde{f}\|_{L^p(\lambda)} \leq c'_{d,p}\, g(W_{N+1})$$

*where $\tilde{f}$ is constructed for the parameters $N$, $p$ and $d$, and where for all $W \geq 1$,*

$$g(W) = \begin{cases} W^{-1/d} & \text{if } (d-1)p < d, \\ W^{-\frac{1}{p(d-1)}} & \text{if } (d-1)p > d, \\ W^{-1/d} \log W & \text{if } (d-1)p = d. \end{cases}$$

*Proof.* Again, we distinguish three cases depending on the values of $p$ and $d$.

- Suppose that $p(d-1) < d$: if $p = 1$, $\frac{2^{d-1}}{K} = \frac{1}{2} < 1$; if $p > 1$, since $\frac{1}{p-1} > d-1, \beta > d-1$ and $\frac{2^{d-1}}{K} = 2^{d-1-\beta} < 1$. Thus, in both cases $\frac{2^{d-1}}{K} < 1$ and for all $N \geq 1$,

$$W_N \leq 2^{Nd} \left(\frac{4d(d+1)^2}{1 - 2^{d-1-\beta}}\right) =: 2^{Nd} c''_{d,p}.$$

Writing the inequality for $N + 1$, we obtain

$$W_{N+1} \leq 2^{Nd} 2^d c''_{d,p}.$$

That is: $2^{-N} \leq 2 \left(\frac{c''_{d,p}}{W_{N+1}}\right)^{1/d}$. Combined with (32), this provides

$$\|f - \tilde{f}\|_{L^p(\lambda)} \leq 2c_{d,p} \left(\frac{c''_{d,p}}{W_{N+1}}\right)^{1/d} = d_{d,p} W_{N+1}^{-1/d},$$

for $d_{d,p} = 2c_{d,p}(c''_{d,p})^{1/d}$ and all $N \in \mathbb{N}^*$.

- If $p(d-1) > d$, then $\beta < d-1$ and $\frac{2^{d-1}}{K} = 2^{d-1-\beta} > 1$. Thus, reminding the definition of $l$ in (28), we have for all $N \geq 1$

$$W_N \leq 2^{Nd}2^{(d-1-\beta)(l+1)}\left(\frac{4d(d+1)^2}{2^{d-1-\beta}-1}\right) \leq 2^{Nd}2^{(d-1-\beta)(N/\beta+2)}\left(\frac{4d(d+1)^2}{2^{d-1-\beta}-1}\right)$$

$$= 2^{N(d+(d-1)/\beta-1)}\left(\frac{4d(d+1)^2 2^{2(d-1-\beta)}}{2^{d-1-\beta}-1}\right) =: 2^{N(1+\frac{1}{\beta})(d-1)}c''_{d,p},$$

for a different constant $c''_{d,p}$. Writing again this inequality for $N+1$, we obtain

$$W_{N+1} \leq c''_{d,p}2^{(1+\frac{1}{\beta})(d-1)}\, 2^{N(1+\frac{1}{\beta})(d-1)},$$

which we can write $2^{-N(1+\frac{1}{\beta})} \leq 2^{(1+\frac{1}{\beta})}\left(\frac{c''_{d,p}}{W_{N+1}}\right)^{\frac{1}{d-1}}$. This provides

$$2^{-N\frac{(1+1/\beta)}{p}} \leq 2^{\frac{(1+1/\beta)}{p}}\left(\frac{c''_{d,p}}{W_{N+1}}\right)^{\frac{1}{p(d-1)}}.$$

Therefore, using (32), we obtain

$$\|f - \tilde{f}\|_{L^p(\lambda)} \leq c_{d,p}2^{\frac{(1+1/\beta)}{p}}\left(\frac{c''_{d,p}}{W_{N+1}}\right)^{\frac{1}{p(d-1)}} = d'_{d,p}W_{N+1}^{-\frac{1}{p(d-1)}},$$

for $d'_{d,p} = c_{d,p}\, 2^{\frac{(1+1/\beta)}{p}}\, (c''_{d,p})^{\frac{1}{p(d-1)}}$ and all $N \in \mathbb{N}^*$.

- If $p(d-1) = d$, then $\beta = d-1$ and $\frac{2^{d-1}}{K} = 1$. Thus, reminding the definition of $l$ in (28), we have for all $N \geq 1$

$$W_N = 2^{Nd+2}d(d+1)^2(l+1) \leq 2^{Nd+2}d(d+1)^2\left(\frac{N}{\beta}+2\right)$$

$$= 2^{Nd}\left(\frac{N}{\beta}+2\right)\left(4d(d+1)^2\right) =: 2^{Nd}\left(\frac{N}{d-1}+2\right)c''_{d,p}$$

$$\leq 2^{d(d-1)\left(\frac{N}{d-1}+2\right)}\left(\frac{N}{d-1}+2\right)c''_{d,p}$$

$$= \exp\left(d(d-1)\left(\frac{N}{d-1}+2\right)\log 2\right)\left(\frac{N}{d-1}+2\right)c''_{d,p} \tag{39}$$

where $c''_{d,p} = 4d(d+1)^2$. Setting

$$\tilde{W}_N := \frac{d(d-1)W_N\log 2}{c''_{d,p}} \qquad \text{and} \qquad \tilde{N} := d(d-1)\left(\frac{N}{d-1}+2\right)\log 2,$$

we can rewrite (39) as:
$$\tilde{W}_N \leq \tilde{N}\exp(\tilde{N}). \tag{40}$$

Since $d \geq 2$, $c''_{d,p} > 0$, $(W_N)_{N\in\mathbb{N}^*}$ is non-decreasing and $\lim_{N\to+\infty} W_N = +\infty$, there exists $W'_{min}$ such that, for all $N$ satisfying $W_N \geq W'_{min}$, we have the following:

$$\begin{cases} \log(\tilde{W}_N) > 1 \\ \log(\tilde{W}_{N+1}) > 2\,\log(2)\,d(d-1) \\ \frac{\log(\tilde{W}_{N+1})}{d\log(2)} - \frac{\log\log(\tilde{W}_{N+1})}{d\log(2)} - 2(d-1) > \frac{1}{p\log(2)} \\ \log W_{N+1} \geq \log\left(\frac{d(d-1)\log 2}{c''_{d,p}}\right). \end{cases} \tag{41}$$

These inequalities will be used latter in the proof and, from now on, we always consider $N$ such that $W_N \geq W'_{min}$.

Let us first show by contradiction that, for all $N$ satisfying $W_N \geq W'_{min}$, (40) implies that

$$\tilde{N} \geq \log \tilde{W}_N - \log\log\tilde{W}_N. \tag{42}$$

Indeed, if the latter does not hold

$$\tilde{N} < \log \tilde{W}_N - \log \log \tilde{W}_N,$$

$$\exp(\tilde{N}) < \frac{\tilde{W}_N}{\log \tilde{W}_N},$$

and therefore, multiplying the two inequalities, since (41) implies that $\tilde{W}_N > 0$, $\log \tilde{W}_N > 0$ and $\log(\log(\tilde{W}_N)) > 0$,

$$\tilde{N} \exp(\tilde{N}) < \tilde{W}_N.$$

The latter being in contradiction with (40), we have proved that, for all $N$ satisfying $W_N \geq W'_{min}$, (42) holds. Using the definition of $\tilde{N}$, we deduce

$$
\begin{aligned}
N &\geq \left( \frac{\log \tilde{W}_N - \log \log \tilde{W}_N}{d(d-1)\log(2)} - 2 \right)(d-1) \\
&= \frac{\log(\tilde{W}_N)}{d\log(2)} - \frac{\log \log \tilde{W}_N}{d\log(2)} + c,
\end{aligned}
$$

for the constant $c = -2(d-1) < 0$. Since $(W_N)_{N \in \mathbb{N}}$ is non-decreasing, for all $N$ satisfying $W_N \geq W'_{min}$, $W_{N+1} \geq W'_{min}$ and the inequality also holds for $N+1$. That is

$$N + 1 \geq \frac{\log(\tilde{W}_{N+1})}{d\log(2)} - \frac{\log \log \tilde{W}_{N+1}}{d\log(2)} + c. \tag{43}$$

Using (32), we obtain:

$$\|f - \tilde{f}\|_{L^p(\lambda)} \leq c_{d,p} N^{\frac{1}{p}} 2^{-N} \leq 2 c_{d,p} (N+1)^{\frac{1}{p}} 2^{-(N+1)}.$$

Since, for $t > \frac{1}{p\log(2)}$, the function $t \longmapsto t^{\frac{1}{p}} 2^{-t}$ is non-increasing, using (43) and (41) and the fact that $-\frac{\log \log \tilde{W}_{N+1}}{d\log(2)} + c < 0$, we obtain

$$
\begin{aligned}
\|f - \tilde{f}\|_{L^p(\lambda)} &\leq 2 c_{d,p} \left( \frac{\log(\tilde{W}_{N+1})}{d\log(2)} \right)^{\frac{1}{p}} 2^{-\frac{\log(\tilde{W}_{N+1})}{d\log(2)}} 2^{\frac{\log \log \tilde{W}_{N+1}}{d\log(2)}} 2^{-c}, \\
&= \left( \frac{2^{1-c} c_{d,p}}{(d\log(2))^{1/p}} \right) (\log \tilde{W}_{N+1})^{\frac{1}{p} + \frac{1}{d}} \tilde{W}_{N+1}^{-\frac{1}{d}} \\
&= \left( \frac{2^{1-c} c_{d,p}}{(d\log(2))^{1/p}} \right) \tilde{W}_{N+1}^{-\frac{1}{d}} \log \tilde{W}_{N+1},
\end{aligned}
$$

since $p(d-1) = d$ implies $\frac{1}{p} + \frac{1}{d} = 1$. Finally, using the definition of $\tilde{W}_N$ and (41), we obtain

$$\|f - \tilde{f}\|_{L^p(\lambda)} \leq d''_{d,p} W_{N+1}^{-\frac{1}{d}} \log W_{N+1},$$

for the constant $d''_{d,p} = 2 \left( \frac{2^{1-c} c_{d,p}}{(d\log(2))^{1/p}} \right) \left( \frac{d(d-1)\log 2}{c''_{d,p}} \right)^{-1/d}$ and all $N \in \mathbb{N}^*$ such that $W_N \geq W'_{min}$. Notice $d''_{d,p}$ only depends on $d$ and $p$.

Taking $c'_{d,p} = \max(d_{d,p}, d'_{d,p}, d''_{d,p})$ provides the announced statement. □

**Proof of Proposition 6.** Take $W_{min} = \max(W'_{min}, W_1)$ and $c = c'_{d,p}$, where $W'_{min}$ and $c'_{d,p}$ are from Lemma 9 and $W_1$ is defined in (38). Let $W \geq W_{min}$, there exists $N \in \mathbb{N}^*$ such that

$$W_N \leq W < W_{N+1}.$$

Consider the architecture $\mathcal{A}$ with $W$ weights, as in Proposition 10, which allows to represent piecewise-constant functions with less than $\frac{W}{2(d+1)^2}$ cubic pieces. It can represent piecewise-constant functions with $\frac{W_N}{2(d+1)^2}$ pieces.

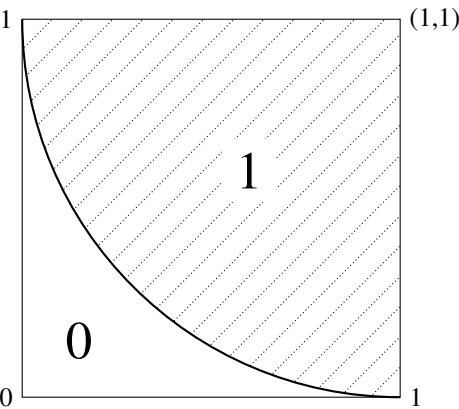

Figure 3: The set $\mathcal{C}$, the set $\partial\mathcal{C} \cap (0,1)^2$ and the indicator function $f$.

For any $f \in \mathcal{M}^d$, the function $\tilde{f}$ obtained for the parameter $N$ is a piecewise-constant function with at most $\frac{W_N}{2(d+1)^2}$ pieces, therefore we have $\tilde{f} \in H_\mathcal{A}$ and, according to Lemma 9, $\tilde{f}$ satisfies

$$\|f - \tilde{f}\|_{L^p(\lambda)} \le c'_{d,p} g(W_{N+1}).$$

Moreover, since $g$ is non-increasing, we have using $c = c'_{d,p}$

$$\|f - \tilde{f}\|_{L^p(\lambda)} \le c\, g(W).$$

Therefore, for any $f \in \mathcal{M}^d$,

$$\inf_{g \in H_\mathcal{A}} \|f - g\|_{L^p(\lambda)} \le c\, g(W)$$

and so does the supremum over $f$ in $\mathcal{M}^d$.

This concludes the proof of Proposition 6.

### E.2 Proof of Proposition 4

**Step 1: we prove the result in dimension $d = 2$.**

We consider the closed disk of radius 1, centered at $(1,1)$,

$$\mathcal{C} = \left\{ x \in \mathbb{R}^2 : \sum_{i=1}^{2} (x_i - 1)^2 \le 1 \right\}.$$

The intersection between $(0,1)^2$ and the topological boundary $\partial\mathcal{C}$ of $\mathcal{C}$ is the quarter of circle:

$$\partial\mathcal{C} \cap (0,1)^2 = \left\{ x \in (0,1)^2 : \sum_{i=1}^{2} (x_i - 1)^2 = 1 \right\}.$$

We denote by $f : [0,1]^2 \to \{0,1\}$ the indicator function of the set $\mathcal{C} \cap [0,1]^2$. The set $\mathcal{C} \cap [0,1]^2$, the set $\partial\mathcal{C} \cap (0,1)^2$ and the function $f$ are represented on Figure 3.

Since no point in $\mathcal{C}^c \cap [0,1]^2$ has all its coordinates strictly larger than those of a point in $\mathcal{C}$, we have $f \in \mathcal{M}^2$ (monotonic functions of 2 variables). We consider an arbitrary neural network architecture $\mathcal{A}$ and $g \in H_\mathcal{A}$.

Let $W \ge 1$ be the number of weights in the architecture $\mathcal{A}$. As is well known for Heaviside neural networks, there exist $K \in \mathbb{N}$ with $K \le 2^W$, reals $\alpha_j$ and polygons $A_j \subset [0,1]^2$, for $j \in \{1, \ldots, K\}$, such that for all $x \in [0,1]^2$

$$g(x) = \sum_{j=1}^{K} \alpha_j \mathbb{1}_{A_j}(x).$$

Moreover, $(A_j)_{1 \le j \le K}$ form a partition of $[0,1]^2$.

The proof relies on the fact (proved afterwards) that, if $\|f - g\|_\infty < \frac{1}{2}$ then $\partial \mathcal{C} \cap (0,1)^2$ is finite. The latter being false, we conclude that $\|f - g\|_\infty \ge \frac{1}{2}$.

Assume from now on that $\|f - g\|_\infty < \frac{1}{2}$. This implies that $g > \frac{1}{2}$ on $\mathcal{C}$, and $g < \frac{1}{2}$ elsewhere. Let us first show that we then have

$$\partial \mathcal{C} \cap (0,1)^2 \; \subset \; \bigcup_{j=1}^{K} \partial A_j.$$

Indeed, if the latter were not true, then there would exist $x \in \partial \mathcal{C} \cap (0,1)^2$ and $j \in \{1, \dots, K\}$ such that $x \in \mathring{A}_j$. Since $\mathcal{C}$ is closed, $x \in \mathcal{C}$. Let $\epsilon > 0$ be such that $B(x, \epsilon) \subset \mathring{A}_j$. We have $B(x, \epsilon) \not\subset \mathcal{C}$ (otherwise, $x$ belongs to the interior of $\mathcal{C}$ which contradicts $x \in \partial \mathcal{C}$). Thus there exists $z \in B(x, \epsilon) \backslash \mathcal{C}$. Since $g > \frac{1}{2}$ on $\mathcal{C}$, and $g < \frac{1}{2}$ elsewhere, we have

$$g(z) < \frac{1}{2} < g(x).$$

This is not possible since $x, z \in \mathring{A}_j$ and $g$ is constant on $A_j$. This concludes the proof of the following fact: if $\|f - g\|_\infty < \frac{1}{2}$ then $\partial \mathcal{C} \cap (0,1)^2 \subset \bigcup_{1 \le j \le K} \partial A_j$.

Since the $A_j$ are polygons (recall that we work in dimension 2), their boundaries are finite unions of closed line segments. Then $\partial \mathcal{C} \cap (0,1)^2$ is included in a finite union of closed line segments which we denote $S_m$, for $m \in \{1, \dots, M\}$. The reader may already see that this is in contradiction with the fact that $\partial \mathcal{C} \cap (0,1)^2$ is a quarter circle. To detail this argument and complete the announced proof, we show that $\partial \mathcal{C} \cap (0,1)^2 \subset \bigcup_{m=1}^{M} S_m$ implies that $\partial \mathcal{C} \cap (0,1)^2$ is finite.

To do so, since when $\partial \mathcal{C} \cap (0,1)^2 \subset \bigcup_{m=1}^{M} S_m$ we have

$$\bigcup_{m=1}^{M} \left( \partial \mathcal{C} \cap (0,1)^2 \cap S_m \right) = \partial \mathcal{C} \cap (0,1)^2,$$

it suffices to prove that the intersection of any closed line segment $S$ with $\partial \mathcal{C} \cap (0,1)^2$ contains at most 2 points.

Denote by $S$ a closed line segment: $\mathcal{C}$ and $S$ are convex and hence connected, thus $\mathcal{C} \cap S$ is either empty, a singleton or a line segment, as a connected compact subset of $S$. If it is empty, then *a fortiori*, $\partial \mathcal{C} \cap (0,1)^2 \cap S = \emptyset$. If it is not, denote by $y$ and $z$ its extremities (assuming $z = y$ in the case of a singleton). By strict convexity of the function $x \mapsto \sum_{i=1}^{2} (x_i - 1)^2$, the open line segment $(y, z)$ is included in $\mathring{\mathcal{C}}$ ( $(y, z) = \emptyset$ in the case of a singleton), hence

$$\partial \mathcal{C} \cap (0,1)^2 \cap S \; \subset \; [y, z] \backslash \mathring{\mathcal{C}} \; \subset \; \{y, z\}.$$

In any case, we have $|\partial \mathcal{C} \cap (0,1)^2 \cap S| \le 2$.

This concludes the proof of the fact: if $\|f - g\|_\infty < \frac{1}{2}$ then $\partial \mathcal{C} \cap (0,1)^2$ is finite and concludes the proof in the case $d = 2$.

**Step 2: we prove the result in any dimension $d \ge 2$,** by a reduction to dimension 2.

We define

$$\mathcal{C} = \left\{ x \in \mathbb{R}^d : \sum_{i=1}^{d} (x_i - 1)^2 \le 1 \right\},$$

and the function $f : [0,1]^d \to \mathbb{R}$ by

$$f(x_1, \dots, x_d) = \mathbb{1}_{(x_1, \dots, x_d) \in \mathcal{C}}.$$

Consider an arbitrary neural network architecture $\mathcal{A}$ and $g \in H_{\mathcal{A}}$. That is, $g$ can be represented by a Heaviside neural network with $d$ input neurons. Note that

$$\sup_{x_1, x_2, x_3 \ldots, x_d \in [0,1]} |f(x_1, x_2, x_3 \ldots, x_d) - g(x_1, x_2, x_3 \ldots, x_d)|$$

$$\geq \sup_{x_1, x_2 \in [0,1]} |f(x_1, x_2, 1 \ldots, 1) - g(x_1, x_2, 1 \ldots, 1)|$$

$$\geq \frac{1}{2},$$

where the last inequality is by the result of Step 1, since $(x_1, x_2) \in [0,1]^2 \mapsto f(x_1, x_2, 1 \ldots, 1)$ is the indicator function of Step 1, and $(x_1, x_2) \in [0,1]^2 \mapsto g(x_1, x_2, 1 \ldots, 1)$ can be represented by a Heaviside neural network with 2 input neurons. This concludes the proof.

**Remark.** Note from the above proof that, though we only stated the impossibility result for piecewise-constant activation functions, an analogous statement in fact holds more generally for piecewise-affine activation functions.

## F  Barron space

In Section 5 we mentioned that the Barron space introduced in [Bar93] is one among several examples for which approximation theory provides ready-to-use lower bounds on the packing number. This space has received renewed attention recently in the deep learning community, in particular because its "size" is sufficiently small to avoid approximation rates depending exponentially on the input dimension $d$. Next we detail how to apply Corollary 1 in this case.

**Definition of the Barron space.**   We start by introducing the Barron space, as defined in [PV21]. Let $d \in \mathbb{N}^*$. For any constant $C > 0$, the Barron space $B_d(C)$ is the set of all functions $f : [0,1]^d \to [0,1]$ for which there exist a measurable function $F : \mathbb{R}^d \to \mathbb{C}$ and some $c \in [-C, C]$ such that, for all $x \in [0,1]^d$,

$$f(x) = c + \int_{\mathbb{R}^d} (e^{ix \cdot \xi} - 1) F(\xi) \mathrm{d}\xi \qquad \text{and} \qquad \int_{\mathbb{R}^d} \|\xi\|_2 |F(\xi)| \mathrm{d}\xi \leq C,$$

where $x \cdot \xi$ denotes the standard scalar product in between $x$ and $\xi$.

**Known lower bound on the packing number.**   Petersen and Voigtlaender [PV21] showed a tight lower bound on the log packing number in $L^p(\lambda, [0,1]^d)$ norm, which we recall below.

**Proposition 11** (Proposition 4.6 in [PV21])**.** *Let $1 \leq p \leq +\infty$. There exist constants $\varepsilon_0, c_0 > 0$ depending only on $d$ and $C$ such that for any $\varepsilon \leq \varepsilon_0$,*

$$\log M(\varepsilon, B_d(C), \|\cdot\|_{L^p}) \geq c_0 \varepsilon^{-1/(\frac{1}{2} + \frac{1}{d})}. \tag{44}$$

**Consequence on the approximation rate by piecewise-polynomial neural networks.**   Plugging the lower bound of Proposition 11 in Corollary 1, we obtain the following lower bound on the approximation error of the Barron space by piecewise-polynomial neural networks.

**Proposition 12.** *Let $1 \leq p < +\infty$, $d \geq 1$. Let $\sigma : \mathbb{R} \to \mathbb{R}$ be a piecewise-polynomial function on $K \geq 2$ pieces, with maximal degree $\nu \in \mathbb{N}$. Consider the Barron space $B_d(C)$ defined above, with $C > 0$. There exist positive constants $c_1, c_2, c_3, W_{\min}$ depending only on $d$, $p$, $C$, $K$ and $\nu$ such that, for any architecture $\mathcal{A}$ of depth $L \geq 1$ with $W \geq W_{\min}$ weights, and for the activation $\sigma$, the set $H_{\mathcal{A}}$ (cf. Section 1) satisfies*

$$\sup_{f \in B_d(C)} \inf_{g \in H_{\mathcal{A}}} \|f - g\|_{L^p(\lambda)} \geq \begin{cases} c_1 W^{-1 - \frac{2}{d}} \log^{-1 - \frac{2}{d}}(W) & \text{if } \nu \geq 2, \\ c_2 (LW)^{-\frac{1}{2} - \frac{1}{d}} \log^{-\frac{3}{2} - \frac{3}{d}}(W) & \text{if } \nu = 1, \\ c_3 W^{-\frac{1}{2} - \frac{1}{d}} \log^{-\frac{3}{2} - \frac{3}{d}}(W) & \text{if } \nu = 0. \end{cases} \tag{45}$$