# OpenReview forum: "A general approximation lower bound in $L^p$ norm, with applications to feed-forward neural networks"
_NeurIPS.cc/2022/Conference — NeurIPS 2022 Accept_

### Official Review · Reviewer_GfMK · 2022-07-04

**Rating:** 6
**Confidence:** 3
**Soundness:** 3 good
**Presentation:** 4 excellent
**Contribution:** 2 fair

**Summary:**

This paper provides a general approximation lower bound result in the $L^p$ norm, and its applications to the neural-network setting. This result is built upon a probability result in [1], which relates the packing number in the $L^p$ norm of a function set $G$ and the fat-shattering dimension of $G$. Given this result, one can derive the approximation in the $L^p$ norm in a relatively straightforward way. Combining existing results on the pseudo-dimension of neural networks, the authors apply the result of $L^p$ norm approximation between two function classes to the neural-network setting.

The authors further apply the neural-network approximation result to the Hölder-ball and monotonic-function settings, to demonstrate the similarity and difference of approximation in the sup norm and the $L^p$ norm.

[1]S. Mendelson. Rademacher averages and phase transitions in glivenko-cantelli classes. IEEE Transactions on Information Theory, 48, 2002.

**Questions:**

Could the authors explain what practical ramifications this new approximation in $L^p$ norm result can bring given the known sup norm approximation results?

**Limitations:**

No major concens. One notational suggestion: I believe that the constants $c_1$ in Equations (10) and (11) are different. You might consider changing $c_1$ in Equation (11) to $c_3$.

**Strengths And Weaknesses:**

Strengths:
1. Studying the approximation in the $L^p$ norm is a theoretically interesting problem in its own. Though the approximation result does not appear very surprising given the existing results provided by the authors, the approximation in the $L^p$ norm is a necessary complement to our current understanding of the approximation problem.
2. This paper is well-written. The results are technical, however, the authors have presented the paper in a quite accessible way.

Weaknesses:
Moving from sup norm to $L^p$ norm is an interesting mathematical and theoretical problem. However, it is not obvious how in practice these two norms can make a difference.

---

> ### Author Response · Authors · 2022-08-01
> **Response to Reviewer GfMK**
>
> Thank you for your positive feedback, and for acknowledging the need to complement existing theoretical results in sup norm with results in $L^p$ norm. We would also like to stress that the $L^p$ norm is in fact more practical than the sup norm, since the $L^2$ norm is ubiquitous in regression tasks (square loss).
>
> **Why can the two norms make a difference in practice?**
> The sup norm and the $L^p$ norm can make a difference in practice when one of the two functions at hand, say $f$, is discontinuous. In this case, $g$ can be terribly bad at approximating $f$ over the whole input domain (which translates into a large sup norm $\Vert f-g\Vert_{\infty}$), while being relatively close to $f$ on most of the input domain except near discontinuities (which translates into a small $L^p(\mu)$ norm $\Vert f-g\Vert_{L^p(\mu)}$). Discontinuous functions are not rare in engineering applications (cf., e.g., a complex physical system with phase transitions, or a complex computer code based on some tabular data), and we studied one special case: monotonic functions (which can be discontinuous). We will summarize these motivations after line 38 in the introduction.
>
> **Practical ramifications of our results?**
> Suppose an engineer wants to design a piecewise-polynomial feed-forward neural network to approximate some black-box function $f$, with as small a network as possible (for energy-consumption / embeddability reasons). How many weights $W$ should be considered to guarantee some approximation accuracy $\varepsilon>0$?
> The results of Section 3 show that, to approximate smooth functions $f$, the minimum number $W$ of weights does not depend on whether $f$ should be approximated on the whole input domain (approximation in sup norm) or on most of the input domain (approximation in $L^p(\mu)$ norm). However, the results of Section 4 indicate that if $f$ is monotonic, then one should choose a large number of weights if worst-case errors are very problematic (as, e.g., in safety-critical engineering), while a smaller number of weights should be sufficient if large errors are tolerated on small regions of the input domain (as, e.g., in marketing applications).

---

> > ### Comment · Reviewer_GfMK · 2022-08-03
> > **Post-rebuttal review**
> >
> > I have read the response. The practical examples provided appear artificial and reverse-engineering the theoretical result. However, I believe that the theoretical contribution of this paper should be acknowledged, so I will keep my rating.

---

### Official Review · Reviewer_voDQ · 2022-07-06

**Rating:** 4
**Confidence:** 3
**Soundness:** 3 good
**Presentation:** 3 good
**Contribution:** 2 fair

**Summary:**

This paper proves a new lower bound on the worst-case approximation error of functions in a set $F$ using functions in another set $G$. The bound is then instantiated in the specific case where $G$ is a piecewise-polynomial feedforward neural network. The approximation is considered in $L^p(\mu)$ norm for any $p \geq 1$.

**Questions:**

If possible, I would consider clarifying whether there would be any practical application of the lower bounds proved in the paper.

A small correction: in line 132, "finite" -> "compact" (or "finite-length", or something else)

**Limitations:**

I don't see any issues here.

**Strengths And Weaknesses:**

The paper is original, clear, well written. Although I haven't checked all details, the authors appear to have done a good job at carrying out precise arguments.

The significance on this paper is however not clear. The results are very theoretical and there is no evidence that they would be relevant for any practical application.

---

> ### Author Response · Authors · 2022-08-01
> **Response to Reviewer voDQ**
>
> Thank you for your positive comments as well as your question about the practical applications of our theoretical results. As for many other lower bounds, the approximation lower bounds we prove in the paper establish fundamental limits that no practical method can outperform. More precisely, our lower bounds have the following practical implications:
> - Consider an engineer who would like to approximate a black-box function $f$ with a piecewise-polynomial neural network, with a guaranteed approximation error of $\varepsilon>0$. The function $f$ is only known to belong to some set $F$ (e.g., it is the solution of a system of PDEs). As briefly mentioned after Corollary 1, our approximation lower bounds imply lower bounds on the number $W$ of weights that must be considered to guarantee accuracy $\varepsilon$ in the worst case. For instance, an approximation lower bound of, say, $W^{-2/\alpha}$ indicates that the engineer should design a neural network with at least $\varepsilon^{-\alpha/2}$ weights.
> - If one finds a feed-forward neural network architecture that provably reaches some approximation lower bound (as in Propositions 2 and 6), then we cannot expect much improvement in terms of worst-case approximation by looking for other feed-forward neural network architectures, unless we consider more weights or layers. This can be useful in the process of designing optimal architectures---though a low worst-case approximation error is just one among several criteria that a good neural network should satisfy.
>
> We would like to add that, probably for the same reasons, several papers in the previous NeurIPS editions tried to contribute to better characterize the expressivity of neural networks, in a (challenging) attempt to understand but also guide neural network practice. Some references include, in chronological order:
>
> [1] Michael Schmitt. Lower bounds on the complexity of approximating continuous functions by sigmoidal neural networks, NeurIPS 1999.
>
> [2] Olivier Delalleau and Yoshua Bengio. Shallow vs. deep sum-product networks, NeurIPS 2011.
>
> [3] Guido F Montufar, Razvan Pascanu, Kyunghyun Cho, and Yoshua Bengio. On the number of linear regions of deep neural networks, NeurIPS 2014.
>
> [4] Zhou Lu, Hongming Pu, F. Wang, Zhiqiang Hu, and Liwei Wang. The expressive power of neural networks: A view from the width, NeurIPS 2017.
>
> [5] Hongzhou Lin and Stefanie Jegelka. ResNet with one-neuron hidden layers is a universal approximator, NeurIPS 2018.
>
> [6] Minshuo Chen, Haoming Jiang, Wenjing Liao, and Tuo Zhao. Efficient approximation of deep relu networks for functions on low dimensional manifolds, NeurIPS 2019.
>
> [7] Dmitry Yarotsky and Anton Zhevnerchuk. The phase diagram of approximation rates for deep neural networks, NeuRIPS 2020.
>
> [8] Christoph Hertrich, Amitabh Basu, Marco Di Summa, and Martin Skutella. Towards lower bounds on the depth of ReLU neural networks, NeurIPS 2021.

---

> > ### Comment · Reviewer_voDQ · 2022-08-09
> > **Thanks**
> >
> > Thanks for your reply! I am not totally convinced by the practicality of the suggested applications, so I will keep my score for now and possibly discuss the matter with the other reviewers.

---

### Official Review · Reviewer_S7Xf · 2022-07-06

**Rating:** 8
**Confidence:** 3
**Soundness:** 4 excellent
**Presentation:** 4 excellent
**Contribution:** 4 excellent

**Summary:**

The paper considers the problem of approximating functions in a family F by functions in another family G. More precisely, they consider the lower bound of the best approximation error $\sup_{f \in F}\inf_{g\in G} \Vert f - g\Vert_{L_p(X,\mu)}$ for real-valued functions, where $X$ is subset of $\mathbb{R}^n$ and $\mu$ is a probability measure on $X$.

The paper obtains a lower bound for bounded functions in terms of an inequality constraint which is expressed in terms of the packing number of $F$ and the fat-shattering dimension of $G$. The paper then uses this result to derive the lower bounds for the approximation of Hölder-continuous functions and multivariate monotonic functions by piecewise polynomial feedforward neural networks.

The main technical argument is heavily built upon a previous result by Mendelson. Still, it is nice work to employ it to solve the problem here, which was an open problem and believed to require new machinery other than the VC dimension theory, and proving a lower bound of the packing number of different families of functions is also nontrivial.


**Questions:**

I do not have any major questions. It would be good if the authors could give some comments on the tightness of their lower bounds.

The following are some minor points on writing:
- Line 25: “functions f in F can” -> “can functions f in F”
- Line 29: sup norm -> supremum norm (or just write L_infty norm if space is an issue). This has multiple occurrences throughout the paper.
- Line 29: the definition of sup norm is incorrect - it should be the essential supremum instead of supremum. The definition in Line 97 is correct though.
- Lines 49 and 53: how is the smoothness of a function defined? I don’t see this in the ‘Main definitions and notation’ section, either. Do you mean the Hölder exponent?
- Line 53: prove -> proves
- Line 61: add a comma before “where”
- Line 102: the usual notation of the packing number is like $M(F, \Vert \cdot \Vert, \epsilon)$, where the parameters are in a different order.
- Line 137 and footnote on the same page: inequation -> inequality; inequation is a word that is rarely used or used only when the direction of the inequality is unknown
- Line 158: above corollary -> corollary above
- Line 194: “$\leq \operatorname{fat}\_{\epsilon/32}(G)$” should be “$\leq \operatorname{fat}\_{\gamma}(G)$”
- Line 219: the measure lambda has been defined in Line 97, so the sentence can be omitted.
- Lines 679 and 680: “cancels” - I think the correct English word is “vanishes” if you mean the functions are zero outside the region


**Limitations:**

N.A.

**Strengths And Weaknesses:**

Strengths: Earlier results only concern the approximation in the L_infinity norm and the approximation in the L_p norm was an open problem, which was previously believed not within the reach of the VC dimension theory. This paper solves the open problem with VC dimension theory.

Weaknesses: The lower bound is still quite messy and it is not known how tight it is. The techniques of the core result was largely done in the previous works, but I don’t think this per se should be evaluated against the acceptance of the paper.

---

> ### Author Response · Authors · 2022-08-01
> **Response to Reviewer S7Xf**
>
> We really appreciate your very positive feedback. We would also like to thank you for your detailed reading and your suggestions to improve the writing, which we will take into account.
>
> As for the tightness of our lower bounds: we proved in Sections 3 and 4 that the general lower bound of Corollary 1 is nearly tight when applied to Hölder balls or to monotonic functions. Indeed the resulting lower bounds, which are stated in Propositions 3 and 5, match the upper bounds of Propositions 2 and 6 respectively, up to logarithmic factors. Therefore, Theorem 1 and its Corollary 1 are nearly tight in some special cases.
>
> We should however note that there is no hope that Theorem 1 is tight for all sets $F$ and $G$. It is indeed based on counting quantities (packing number of $F$, fat-shattering dimension of $G$), while other geometrical properties of $F$ and $G$ could be key to quantify the approximation error. However, counting arguments can be sufficient to yield nearly tight lower bounds whenever the geometries of $F$ and $G$ agree, as is the case in Sections 3 and 4, and likely in other settings mentioned in Section 5 (e.g., approximation of the Barron space).

---

> > ### Comment · Reviewer_S7Xf · 2022-08-03
> > **Thanks for the response**
> >
> > I think my question is adequately answered.

---

### Official Review · Reviewer_qRrC · 2022-07-11

**Rating:** 6
**Confidence:** 3
**Soundness:** 3 good
**Presentation:** 3 good
**Contribution:** 3 good

**Summary:**

This paper studies the lower bound of the approximation error with respect to Lp norm. The approximation error means how well functions f can be approximated by functions g_w with parameters w. Besides the general case, the authors also provide results for some special cases where g_w is to a feed-forward neural network, f is Holder balls and monotonic functions. Matching upper bounds of the approximation error for those special cases are also provided to show that these bounds are relatively tight.

**Questions:**

1. In Section 4, the authors study the situation of monotonic functions. I wonder why monotonic functions are special/important in the sense of approximation performance.

2. It is not easy to see the effect of p in most of theorems and corollaries. Maybe the authors can provide some explanation on how different p affects the bound of the approximation error.

3. When considering the feed-forward neural network, the activation functions are limited to piecewise-polynomial ones (e.g., ReLU). Maybe the authors can discuss why piecewise-polynomial activation functions are essential for the current derivation, or, how other types of activation functions (e.g., sigmoid) make the estimation of approximation error difficult.

**Strengths And Weaknesses:**

Strength (dominating): The result and contributions are clearly stated. Besides, this paper has good writing quality and most of the contents are easy to follow.

Weaknesses: Some details need more explanation. See my questions below.

---

> ### Author Response · Authors · 2022-08-01
> **Response to Reviewer qRrC**
>
> Thank you for your positive comments and suggestions for clarification. Please find answers to your three questions below.
>
> 1. The reasons why we studied monotonic functions in Section 4 are twofold. First, they naturally appear in physics or engineering applications. Consider, e.g., an autonomous vehicle. Predicting the number of kilometers that can be traveled, or predicting the braking distance at a given time, given input variables such as the total load, the speed, the drag coefficient, etc, typically correspond to a monotonic input-output relationship. The second reason is that monotonic functions are natural examples of possibly discontinuous functions, which also appear in engineering. With discontinuities, approximation in sup norm can be difficult (see our impossibility result in Proposition 4), but approximation in $L^p$ norm can be feasible, with an approximation rate that we characterize in Propositions 5 and 6 for the case of monotonic functions. We will add one paragraph at the beginning of Section 4 (before line 269) to summarize these two motivations.
>
> 2. In the two applications we considered, the parameter $p$ has either no effect on the approximate rate (for Hölder balls, the right-hand side of (13) does not depend on $p$ except in the multiplicative constant $c_1$), or an important effect on the approximate rate (for monotonic functions, the right-hand side of (14) depends heavily on $p$ through the exponent $\alpha = \max(d,(d-1)p) = (d-1)p$ whenever $p > 1 + 1/(d-1)$, and in particular when $p > 2$). The effect of $p$ thus depends on the set $F$ (the presence or absence of discontinuities play a crucial role).
> We will add one sentence after each of Propositions 3 and 5 to make the dependency on $p$ clearer.
>
> 3. Piecewise-polynomial activation functions are not essential for the current derivation. Indeed, Theorem 1 can also be applied to the case where $G$ corresponds to a neural network with other activation functions such as the sigmoid. In the sigmoid case, the pseudo-dimension is known to be at most of the order of $W^4$ (see the references [1,2] below), which we can use to derive an approximation lower bound similar to that of Corollary 1, but with a different rate in $W$ (the lower bound for sigmoid networks is smaller). What makes the picture more difficult here is that, as of now and to the best of our knowledge, it is not known whether the $\mathcal{O}(W^4)$ VC bound is tight (only a lower bound of the order of $W^2$ is known), so the resulting approximation lower bound could be loose. We will add a comment about sigmoid networks before the last paragraph of Section 5.
>
> [1] Marek Karpinski and Angus Macintyre. Polynomial bounds for VC dimension of sigmoidal and general Pfaffian neural networks. Journal of Computer and System Sciences, 54(1):169–176, 1997.
>
> [2] Martin Anthony and Peter L. Bartlett. Neural network learning: theoretical foundations. Cambridge University Press, Cambridge, 1999.

---

### Author Response · Authors · 2022-08-01
**Summary of reviews and responses**

We would like to thank again all four reviewers for their positive feedback and the time spent reading the paper. Beyond suggestions to improve the writing, the reviewers asked questions:
- about the effect of $p$, the choice of the activation function, and the tightness of our lower bounds;
- about the practical significance of our theoretical results: why studying monotonic functions in Section 4? How relevant are our results for practical applications? What are the practical ramifications of our new results in $L^p$ norm given the known sup norm approximation results?

We would like to sum up our responses about practical significance. Monotonic functions (studied in Section 4) are typical in engineering applications and are natural examples of possibly discontinuous functions, which are hard to approximate on the whole input domain (i.e., in sup norm), but that can be approximated on most of the input domain (i.e., in $L^p$ norm). Our new $L^p$ norm results quantify this phenomenon and shed some light on a classical choice in practice: the $L^2$ norm (regression with the square loss). From an engineering viewpoint, our lower bounds imply constraints on the minimum number $W$ of weights than an engineer should consider to guarantee some approximation accuracy $\varepsilon>0$.

Such results, as well as all other approximation results that appeared in the previous editions of NeurIPS (a list of references can be found in the response to Reviewer voDQ), are little but hopefully useful steps towards certified neural network learning (with a guaranteed approximation accuracy).

---

### Meta-Review · Area_Chair_Jtu3 · 2022-08-26

**Recommendation:** Accept
**Confidence:** Certain

**Metareview:**

The paper provides novel lower bounds on function approximation, relating $L^p$ norm approximation error to combinatorial complexity measures of both the approximating and approximated functions classes. These bounds are instantiated for approximation via piecewise-polynomial neural networks.

There is a consensus among the reviewers that the results of the paper are novel, solve an open problem, and follow from deep technical insights. Consequently, I concur with the majority of reviewers and recommend acceptance of the paper.

**Award:**

No

---

### Decision · Program_Chairs · 2022-09-14

Accept